# Liquid Cloud Optical Properties Retrieval and Associated Uncertainties using Multi-angular and Bispectral Measurements of the Airborne Radiometer OSIRIS

Christian Matar[1,a], Céline Cornet[1], Frédéric Parol[1], Laurent C-Labonnote[1], Frédérique Auriol[1], and Jean-Marc Nicolas[1]

[1] Univ. Lille, CNRS, UMR 8518 - LOA - Laboratoire d'Optique Atmosphérique, F-59000 Lille, France

[a] now at: GRASP SAS, 59260 Lezennes, France

*Correspondence to*: Christian Matar (christian.matar@grasp-sas.com)

**Abstract.** In remote sensing applications, clouds are generally characterized by two properties: cloud optical thickness (COT) and effective radius of water/ice particles ($R_{eff}$), and additionally by geometric properties when specific information is available. Most of the current operational passive remote sensing algorithms use a mono-angular bispectral method to retrieve COT and $R_{eff}$. They are based on pre-computed lookup tables while assuming a homogeneous plane-parallel cloud layer. In this work, we use the formalism of the optimal estimation method, applied to airborne near-infrared high-resolution multi-angular measurements, to retrieve COT and $R_{eff}$, and the corresponding uncertainties related to the measurement errors, to the non-retrieved parameters and the cloud model assumptions. The used measurements were acquired by the airborne radiometer OSIRIS (Observing System Including PolaRization in the Solar Infrared Spectrum), developed by the Laboratoire d'Optique Atmosphérique. It provides multi-angular measurements at tens of meters resolution, very suitable for refining our knowledge of cloud properties and their high spatial variability. OSIRIS is based on the POLDER (POlarization and Directionality of the Earth's Reflectances) concept as a prototype of the future 3MI space instrument planned to be launched on the EUMETSAT-ESA MetOp-SG platform in 2024. The used approach allows the exploitation of all the angular information available for each pixel to overcome the radiance angular effects. More consistent cloud properties with lower uncertainty compared to operational mono-directional retrieval methods (traditional bispectral method) are then obtained. The framework of the optimal estimation method also provides the possibility to estimate uncertainties of different sources. Three types of errors were evaluated: (1) Errors related to measurement uncertainties, which reach 6 and 12% for COT and $R_{eff}$ respectively, (2) errors related to an incorrect estimation of the ancillary data that remain below 0.5%, and (3) errors related to the simplified cloud physical model assuming independent pixel approximation. We show that not considering the in-cloud heterogeneous vertical profiles and the 3D radiative transfer effects leads to an average uncertainty of 5% and 4% for COT and 13% and 9% for $R_{eff}$.

# 1 Introduction

The role and evolution of clouds in the ongoing climate change are still unclear. Their radiative feedback due to temperature rise or due to the indirect effect of aerosols is insufficiently understood and they are known to contribute to the uncertainties in the Earth future climate (IPCC report, 2021). An accurate estimation of cloud properties is therefore very important for constraining climate and meteorological models, improving the accuracy of climate forecasting, and monitoring the cloud cover evolution. The instruments onboard Earth observation satellites allow continuous monitoring of the clouds and aerosols and retrieval of their properties from a regional to a global scale.

The cloud properties are retrieved using the information carried by the measurements of the reflected, emitted, or transmitted radiations by the clouds. Two main optical cloud properties are generally retrieved: the cloud optical thickness (COT) and the effective radius of the water/ice particles forming the cloud ($R_{eff}$). These optical properties, along with the cloud altitude when possible, allow to characterize the clouds at a global scale and help to determine the radiative impacts of clouds along with their cooling and warming effects (Twomey, 1991; Lohmann and Feichter, 2005; Rivoire et al., 2020; Yang et al., 2010). Depending on the available information, various passive remote sensing methods are operationally used for the retrieval of these optical properties. For instance, the infrared split-window technique (Giraud et al., 1997; Inoue, 1985; Parol et al., 1991) uses infrared measurements and is more suitable for optically thin ice clouds (Garnier et al., 2012). The bispectral method (Nakajima and King, 1990) which uses visible and shortwave infrared wavelengths, is more suitable for optically thicker clouds. It is currently used in a lot of operational algorithms, for example by the MODIS radiometer (Platnick et al., 2017). It is also possible to use a combination of multi-angular total and polarized measurements in the visible range, such as POLDER measurements (Deschamps et al., 1994), to retrieve COT and $R_{eff}$ (Bréon and Goloub, 1998; Buriez et al., 1997).

The above-mentioned methods are subject to several sources of error. A moderate perturbation in the retrieved COT and $R_{eff}$ can, for example, cause variations of around 1 to 2 W/m$^2$ in the estimation of cloud radiative forcing (Oreopoulos and Platnick, 2008). The quantification of the retrieval uncertainties of these optical properties is therefore critical. The sources of errors originating from the measurements can be quite well evaluated along the instrument calibration process and are often considered when developing a new algorithm (Sourdeval et al., 2015; Cooper et al., 2003, Platnick et al. 2017) but the errors related to the choice of the cloud model to retrieve the parameters and the assumption made for the radiative transfer simulations should not be overlooked. Currently, computational constraints and lack of information in the measurements force the operational algorithms of cloud products (MODIS, POLDER, and others) to retrieve the cloud optical properties with a simplified 1D-cloud model. In this model, clouds are considered flat between two spatially homogeneous planes in what is known as the plane-parallel and homogeneous (PPH) assumption (Cahalan et al., 1994). Another commonly used assumption is related to the infinite dimension of the PPH cloud and treats each pixel independently without considering the interactions that occur between neighboring homogeneous pixels, known as the independent pixel approximation (IPA) (Cahalan et al., 1994; Marshak et al., 1995b). The effect of these two assumptions can lead to large uncertainties and bias regarding the cloud properties (Marshak et al., 2006b; Seethala and Horváth, 2010) and the aerosol-cloud relationship (Kaufman et al., 2002; Chang and Christopher, 2016).

Considering the spatial variability of the cloud macrophysical and microphysical properties, the errors induced by the use of a homogeneous horizontal and vertical cloud model have been found to depend on the spatial resolution of the observed pixel, the wavelength, and the observation and illumination geometries (Kato and Marshak, 2009; Zhang and Platnick, 2011; Zinner and Mayer, 2006; Davis et al., 1997; Oreopoulos and Davies, 1998; Várnai and Marshak, 2009).

From medium to large-scale observations greater than 1 km (e.g. MODIS: $1\times1$ km$^2$, POLDER: $6\times7$ km$^2$), the PPH approximation poorly represents the cloud variability. The subpixel horizontal heterogeneity and the nonlinear nature of the COT-radiance relationship create the PPH bias that leads to the underestimation of the retrieved COT (Cahalan et al., 1994; Szczap et al., 2000; Cornet et al., 2018). The PPH bias is increasing with pixel size due to the inhomogeneity increase. Using the bispectral method, the COT subpixel heterogeneity induces also an overestimation

bias on the retrieved $R_{eff}$ (Zhang et al., 2012), while this effect appears limited with polarimetric observations (Alexandrov et al., 2012; Cornet et al., 2018). On the contrary, the microphysical subpixel heterogeneity leads to an underestimation of retrieved $R_{eff}$ (Marshak et al., 2006b).

At smaller scales, as considered here, errors due to IPA become more dominant. At this scale, pixels can no longer be considered infinite and independent from their adjacent pixels. Radiative energy pass from one column to the others

depending on the COT gradient. This leads to a decrease in the radiance of pixels with large optical thickness and an increase in the radiance of pixels with small optical thickness, which tends to smooth the radiative field and thus the field of retrieved COT (Marshak et al., 1995a, b). As a result, it can lead to a large underestimation of the retrieved optical thickness (Cornet and Davies, 2008). Adding to these effects, for off-nadir observations, the tilted line of sight crosses different atmospheric columns with variable extinctions and optical properties which tend to additionally

smooth the radiative field (Várnai and Davies, 1999; Kato and Marshak, 2009; Benner and Evans, 2001; Várnai and Marshak, 2003; Fauchez et al. 2018). In the case of fractional cloud fields not examined under nadir observations, the edges of the clouds cause an increase in the radiances for high viewing angles, which in turn, increases the value of the retrieved COT (Várnai and Marshak, 2007), while overestimating the retrieved $R_{eff}$ (Platnick et al., 2003). They are often filtered out of cloud property retrievals, especially under low sun angles (Takahashi et al., 2017; Zhang et al.,

2019). The illumination and shadowing effects, on the contrary, lead to a roughening of the radiative field by increasing or decreasing radiances compared to the prediction of the plane-parallel homogeneous clouds. Their influence in over and under-estimating the cloud droplet size retrievals are documented in several papers (Zhang et al., 2012; Marshak et al., 2006a; Cornet et al., 2005).

The assumption of a vertically homogeneous profile inside the cloud is also questionable. The vertical distribution of

the cloud droplets is important to provide an accurate description of the radiative transfer in the cloud (Chang, 2002) and obtain a more accurate description of the cloud microphysics such as the water content or the droplet number concentration. For simplicity reasons, classical algorithms assume a vertically homogeneous cloud model. However, several studies have shown a dependence between the retrieved effective radius and the SWIR band used. These differences are explained by the non-homogenous cloud vertical profiles and by the different sensitivities of spectral

channels due to wavelength-dependent cloud particle absorption (Platnick, 2000; Zhang et al., 2012). Indeed, the absorption by water droplets being stronger at 3.7 µm, the radiation penetrates less deeply in the cloud than at 2.2 and 1.6 µm. The use of channel 3.7 is therefore expected to lead to retrieving an effective radius that corresponds to a level

in the cloud higher than that of channels 2.2 and 1.6. Considerable vertical variation along the cloud profiles is confirmed by many in-situ studies of droplet size profiles and water content as summarized in (Miles et al., 2000). This

vertical variation in liquid particle size is an important cloud parameter related to the processes of condensation, collision-coalescence, and the appearance of precipitation (Wood, 2005). The diversity of possible vertical profiles is difficult to account for. Saito et al. (2019) propose a method to retrieve it using Empirical Orthogonal Function (EOF) to reduce the degrees of freedom of the droplet size profile.

In operational algorithms, the retrieval of COT and $R_{eff}$ is achieved through pre-computed Look-Up Tables (LUT).

This method can be used to process large databases automatically. Its disadvantage is that a modification of the particle model or any other model parameter requires re-generating all these pre-computed tables. In addition, until recently, the difficulty was to assess the uncertainties of the retrieved cloud properties. Platnick et al. (2017) succeeded to derive the total uncertainties on COT and $R_{eff}$ and to decompose the contribution of uncertainties from measurement errors and several non-retrieved parameters using covariance matrix and Jacobian computations from LUT.

In this paper, we present a method based on the Optimal Estimation Method (Rodgers et al. 2000) to also derive separately each type of uncertainty and apply it to the measurements of the airborne radiometer named Observing System Including PolaRization in the Solar Infrared Spectrum (OSIRIS), which was developed in the Laboratoire d'Optique Atmosphérique (Auriol et al., 2008). OSIRIS is the airborne simulator of the 3MI (Multi-viewing Multi-channel Multi-polarization Imaging) radiometer, planned to be launched on MetOp-SG in 2024. It can measure the

degree of linear polarization from 440 to 2200 nm and has been used onboard the French Falcon 20 environmental research aircraft of Safire, during several airborne campaigns: CHARMEX/ADRIMED (Mallet et al., 2016), CALIOSIRIS and AEROCLO-sA (Formenti et al., 2019).

We couple the multi-angular multi-spectral measurements of OSIRIS with a statistical inversion method to obtain a flexible retrieval process of COT and $R_{eff}$. The exploitation of the additional information on the cloud provided by these

versatile measurements implies the use of a more sophisticated inversion method compared to the LUT. The optimal estimation method (Rodgers, 1976, 2000) has been widely used for applications in cloud remote sensing (Cooper et al., 2003; Poulsen et al., 2012; Sourdeval et al., 2013; Wang et al., 2016). In this method, the Bayesian conditional probability together with a variational iteration method allows the convergence to the physical state which allows the forward model to best fits the measurements. Therefore, it introduces the probability distribution function of solutions

where the retrieved parameter is the most probable, with an ability to extract separate uncertainties of the retrieved parameters (Whalter et Heidinger, 2012).

The aim of this paper is not to give an exhaustive view of the possible errors concerning optical thickness and effective radius retrievals but to simply introduce a method to derive the different sources of uncertainties from a specific case of data acquired during an airborne campaign. Uncertainties due to error measurements, non-retrieved parameters but

also to the assumed forward model are considered. If generalized to several cloudy scenes, the partitioning of the errors can help to understand if and which non-retrieved parameters or forward models need to be optimized to reduce the global uncertainties of the retrieved cloud parameters.

This article is organized as follows. Section 2 describes the basic characteristics of OSIRIS and some essential details
of the campaign CALIOSIRIS-2. In section 3, a detailed description of the retrieval methodology is presented, including the mathematical framework needed to compute the uncertainties of the retrieved cloud properties. In section 4, a case study of a liquid cloud is presented and analyzed. We assessed the magnitude of different types of errors, such as the errors due to measurement noise, the errors linked to the fixed parameters in the simulations, and the errors related to the unrealistic homogeneous cloud assumption. The multi-angular retrievals and uncertainties are compared
with the results obtained by the classical monoangular bispectral retrieval algorithms in section 5. Finally, section 6 gives a summary and some concluding remarks.

## 2 Instrumentation and airborne campaign

We use the new imaging radiometer OSIRIS. We will go through the main characteristics of the instrument and the airborne campaign CALIOSIRIS. More details about OSIRIS can be found in (Auriol et al., 2008).

## 2.1 OSIRIS

OSIRIS (Observing System Including PolaRisation in the Solar Infrared Spectrum) is an extended version of the POLDER radiometer (Deschamps et al., 1994) with multi-spectral and polarization capabilities extended to the near and short-wave infrared. This airborne instrument is a prototype of the future spacecraft 3MI (Marbach et al., 2015) planned to be launched on MetOp-SG in 2024. It consists of two optical sensors, each one with a two-dimensional
array of detectors; one for the visible and near-infrared wavelengths (from 440 to 940 nm) named VIS-NIR (Visible-Near Infrared) and the other one for the near and shortwave infrared wavelengths (from 940 to 2200 nm) named SWIR (Shortwave Infrared). The VIS-NIR detector contains 1392×1040 pixels with a pixel size of 6.45×6.45 $\mu m^2$ while the SWIR contains 320×256 pixels with a pixel size of 30×30 $\mu m^2$. Adding those characteristics to the wide field of view of both heads, at a typical aircraft height of 10 km, the spatial resolution at the ground is 18 m and 58 m for the VIS-
NIR and SWIR respectively. This leads to a swath of about 25×19 km for the visible and 19×15 km for the SWIR. OSIRIS has eight spectral bands in the VIS-NIR and six in the SWIR. Similar to the concept of POLDER, OSIRIS contains a motorized wheel rotating the filters in front of the detectors. The step-by-step motor allows only one filter to intercept the incoming radiation at a particular wavelength. The polarization measurements are conducted using a second rotating wheel of polarizers. Given the sensor exposure and transfer times, the duration of a full lap is about 7
seconds for the VIS-NIR and 4 seconds for the SWIR. Figure 1 shows the spectral response of each channel of OSIRIS. The two channels (1240 and 2200 nm) used in this study are red colored in the figure.

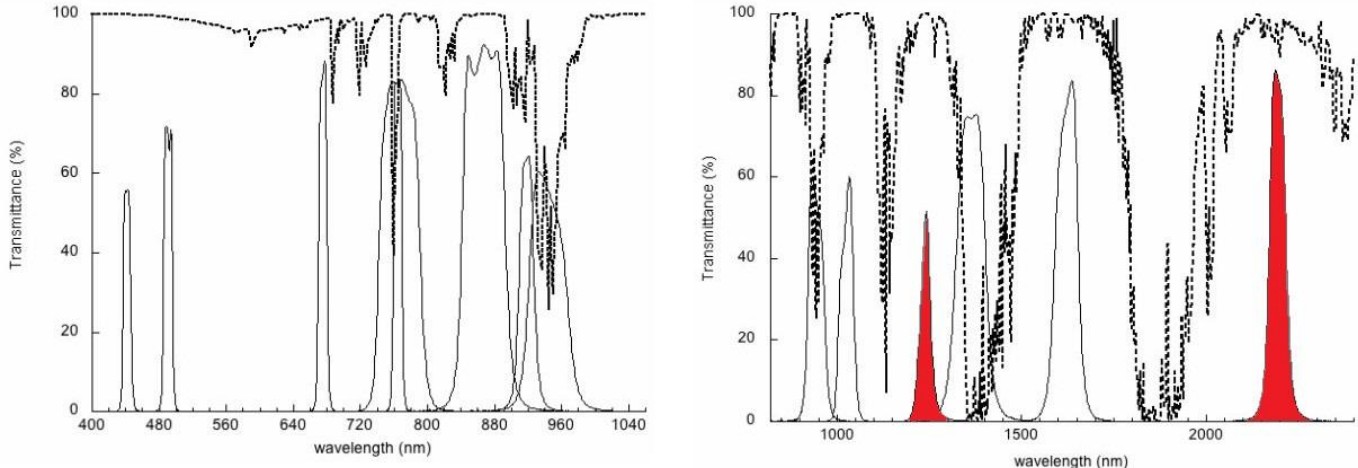

**Figure 1: Spectral wavelengths of VIS-NIR (left) and SWIR (right) spectral response function of OSIRIS optical channels without unit and normalized to unity. The dashed line corresponds to a typical atmospheric transmittance in %. The red-colored channels are used in this study (1240 and 2200 nm).**

OSIRIS is an imaging radiometer with a wide field of view. It has a sensor matrix that allows the acquisition of images with different viewing angles. The same scene can thus be observed several times during successive acquisitions with variable geometries. The largest dimension of the sensor matrix is oriented along-track of the aircraft to increase the number of viewing angles for the same target. For example, when the airplane is flying at a 10 km altitude with a speed of 200 to 250 m/s, the same target on the ground can be seen under 20 different angles for the VIS-NIR and 19 for the SWIR.

## 2.2 Airborne campaign and case study

OSIRIS participated in the airborne campaign CALIOSIRIS in October 2014. It was carried out with the contributions of the French laboratories LOA (Laboratoire d'Optique Atmosphérique) and LATMOS (Laboratoire ATmosphères, Milieux, Observations Spatiales, Paris) and with SAFIRE, the French Facility for Airborne Research. One objective of this campaign was the development of new cloud and aerosol properties retrieval algorithms in anticipation of the future space mission of 3MI intending to improve our knowledge of clouds, aerosols and cloud-aerosol interactions. The data used in this work focuses on a cloudy case over the ocean surface: a marine monolayer cloud that was observed on 24 October 2014 at 11:02 (local time). The aircraft flew at an altitude of 11 km above the Atlantic Ocean facing the French west coast (46.70°,-2.82°, red arrow in Figure2a). The solar zenith angle was equal to 59°. The LNG (lidar aerosols nouvelle generation, Bruneau et al. 2015), a high spectral resolution airborne Lidar at 355nm, was also onboard the Falcon-20 aircraft along with OSIRIS during the airborne campaign. In Figure 2b, the vertical profiles of the backscattered signal measured by the LIDAR-LNG are represented. The red rectangle in Figure 2b corresponds to OSIRIS images and the scene studied in this paper. The LIDAR-LNG detected a monolayer cloud around 5.5 km. In panels (c) and (d), we present colored compositions of total and polarized radiances obtained from three spectral bands of OSIRIS over this cloud scene. One OSIRIS image corresponds to several viewing angles. The zenith angle ranges from about 0° in the center of the image to 55° in the corner of the image. The white concentric contours represent the scattering iso-angles in a step of 10º.

The clouds backscatter total solar radiation more intensely in the cloudbow regions near 140°. The position of the cloudbow peak depends on the wavelength, resulting in the decomposition of the light, slightly visible between the 140° and 150° scattering angle contours. On the polarized image (Figure2d), we observe a stronger directional signature of the signal, characteristic of scattering by spherical droplets showing a cloud bow clearly visible between about 140° and 150°. At larger scattering angles between 150° and 160°, we observe slightly the supernumerary bows whose positions vary with the wavelength, alternating between the red, blue, and green channels. The measured polarized signal for scattering angles smaller than 130° is largely dominated by molecular scattering at 490 nm, hence the blue color. Since the solar zenith angle is 59°, the specular direction corresponds to a scattering angle of 62° in the solar plane (not visible in Figures 2c and 2d) but the ocean wind enlarges the sun glint area, resulting in an enhancement of the radiances between the 70° and 80° scattering iso-contours.

At the time of the CALIOSIRIS campaign in 2014, the polarized channels presented calibration and stray light issues, which make use of the polarized measurements difficult for quantitative retrievals. In addition, the images from the two sensors were not well co-localized. Consequently, for this work, we use two unpolarized channels of the SWIR matrix, one almost non-absorbing (1240 nm) and one absorbing (2200 nm) to have information on optical thickness and effective radius respectively.

In order to use the multi-angular capability of OSIRIS, successive images have to be colocalized. After subtracting the average of similar successive images to remove the angular effects, the colocalization is achieved by minimizing the root mean square difference of the radiances between each pair of successive images for different translations along the line and the column in the second image. The reference image is the central one of the sequence. Images with translations beyond the dimensions of the central image are ignored. Multi-angular radiances at the cloud level correspond in our case to 9 to 13 directions.

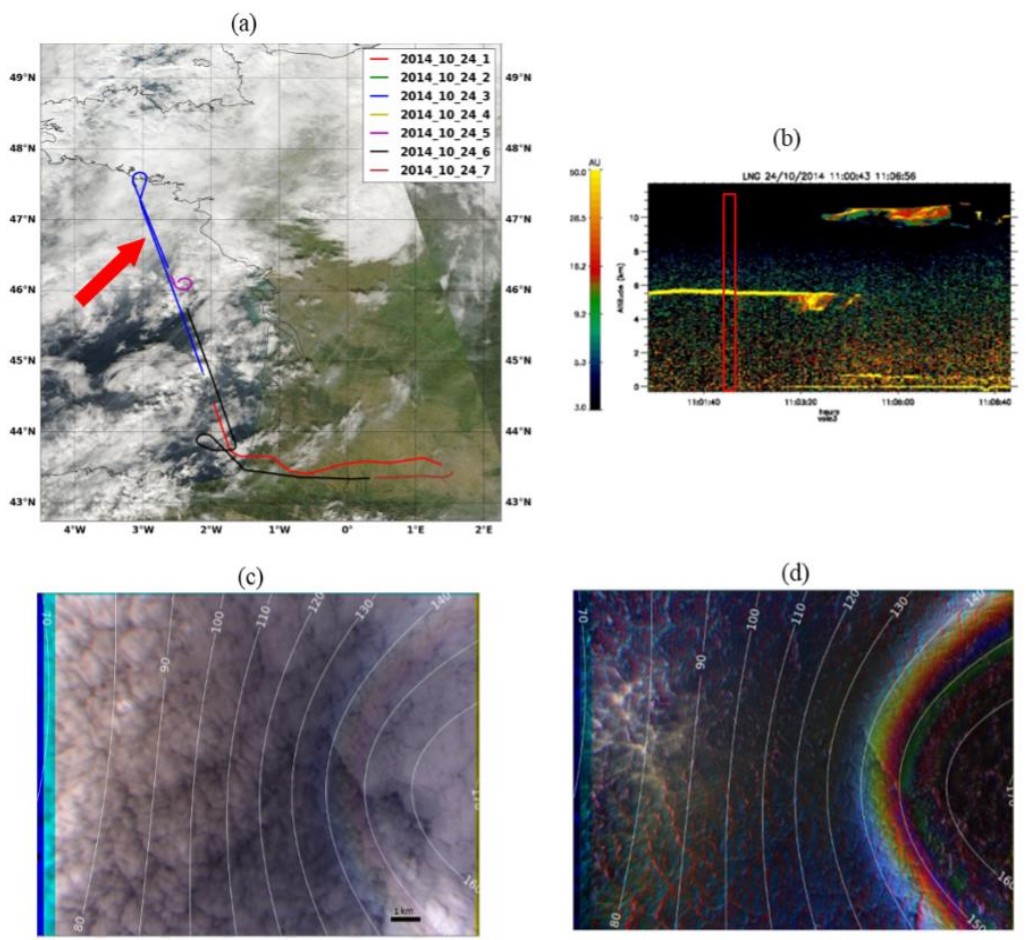


**Figure 2: Studied case on 24 October 2014 at 10:02 UTC (11:02 local time): (a) In blue, airplane trajectory for this day above a MODIS/AQUA true color image. The red arrow corresponds to the studied segment (b) Quicklook of the backscattered signal provided by the LIDAR-LNG around the observed scene. The red rectangle corresponds to the studied scene (c) OSIRIS RGB composite image, obtained from the total radiances at channels 490, 670, and 865 nm. The blue bars on the left-hand side of the images are due to the motion of the airborne between the image acquisitions of the different filters (d) OSIRIS RGB composite image, obtained from the polarized radiances at channels 490, 670, and 865 nm. The white iso-contours in (c) and (d) represent the scattering iso-angles in a 10° step.**

## 3 Retrieval methodology

One of the most robust approaches in cloud property retrievals is the optimal estimation method (OEM). It is increasingly used in satellite measurement inversion (Cooper et al., 2003; Poulsen et al., 2012; Walther and Heindiger, 2012, Sourdeval et al., 2013; Wang et al., 2016). It provides a rigorous mathematical framework to estimate one or more parameters from different measurements. The OEM also characterizes the uncertainty on the retrieved parameters while taking into account the instrument error and the underlying physical model errors. A complete description of the optimal estimation method for atmospheric applications is given by Clive D. Rodgers (Rodgers, 2000). In this book, Rodgers described exhaustively the information content extraction from measurements, the optimization of the inverse problem, and the solutions and error derivations. In the following, we will go through the basics of this method that define the core of our retrieval algorithm.

### 3.1 The formalism of the optimal estimation method

Considering a vector $y$ (of dimension $n_y$) containing the measurements and a state vector $x$ (of dimension $n_x$) containing the unknown properties to be retrieved, these two vectors are connected by the forward model $F$, which can model the complete physics of the measurements to an adequate accuracy. The errors associated with the measurement and the modeling are represented by the error vector $\epsilon$. Eq. (1) states the relationship between these variables.

$$y = F(x) + \epsilon \tag{1}$$

The OEM aims to find the best representation of parameters $x$ that minimizes the difference between simulations $F(x)$
and observations $y$ while considering the linearity of the direct model near the solution. To achieve it, a Bayesian probabilistic approach is applied. Before the measurements, an a priori knowledge of the state vector can be described by a probability density function (PDF) $P(x)$. Once the measurements $y$ have been carried out, this knowledge can be described by the posterior PDF of the state $P(x|y)$, which is a conditional probability (probability of having $x$ given that $y$ is true). The posterior PDF of the state vector can be related to its a priori PDF by Bayes' theorem:

$$P(x|y) = \frac{P(y|x).P(x)}{P(y)} \tag{2}$$

Where $P(y)$ is the PDF of the measurements including the uncertainties and $P(y|x)$ is the PDF of the measurements given that we know the state vector.

In the optimal estimation method, the previous PDFs are represented by Gaussian distributions, assuming that the errors of the measurements, the errors related to the non-retrieved parameters and the errors of the forward model are normally
distributed around a mean value.

Therefore, it can be easily shown that the best estimate of the state vector $x$ corresponds to the minimum of the so-called cost function $J(x)$:

$$J(x) = [y - F(x)]^T S_\epsilon^{-1} [y - F(x)] + [x - x_a]^T S_a^{-1} [x - x_a] \tag{3}$$

The first term of $J(x)$ represents the difference between the measurements and the forward model calculated for a given state vector $x$, weighted by $S_\epsilon$ the covariance matrix associated with the measurement error and the forward model
errors. The second term represents the difference between the state vector $x$ and the a priori state vector $x_a$ weighted by $S_a$ the covariance matrix associated with $x_a$. In line with the cost function, the optimal estimation emerges from a balance between the information carried by the measurement about the state vector and what we already know about it before the measurement. In our case, we do not have a prior estimate of the state vector. The iterations are initiated by a first guess while applying a large $S_a$. The difference between the measurements and the forward model will be the
decisive element in the minimization of the cost function. It will ensure that the estimated cloud properties have the optimal fit with the observed system only.

The minimization is done through the Levenberg-Marquardt approach (Marquardt, 1963; Levenberg, 1944) based on the "Gauss-Newton" iterative method. Assuming the model is nearly linear around a given state vector, each iteration is calculated following Eq. (4):

$$x_{i+1} = x_i + S_{x_i}^{-1} \left[ K_i^T S_\epsilon^{-1} (y - F(x_i)) - S_a^{-1} (x_i - x_a) \right] \tag{4}$$

where $x_i$ is the state vector at the $i_{th}$ iteration, $K_i$ is the sensitivity (or Jacobian) matrix described in Eq. (10) and $S_{x_i}$ is the covariance matrix of the state vector defined in Eq. (5).

$$S_{x_i} = \left[(1 + \gamma)S_a^{-1} + K_i^T S_\epsilon^{-1} K_i\right]^{-1}$$

(5)

The parameter $\gamma$ affects the size of the step at each iteration. If the cost function increases at an iterative step $i$ then $\gamma$ is increased and a new smaller step ($x_{i+1}$) is calculated until the cost function decreases.

The iterative process stops when the simulation fits the measurement (Eq. (6)), named convergence of Type 1 or when the iteration converges (Eq. 7) named convergence of Type 2. The left side of Eq. (6) represents the normalized cost function without taking into account the a priori negligible contribution. When the cost function is smaller than $n_y$ or the normalized cost function ($J/n_y$) less or equal to one, the iterations stop. Eq. (7) deals with the iterative steps and will make sure that the iterations will stop when the difference between two successive steps weighed by $S_x$ is less than

$n_x$. In other words, when further changes in the state vector have small to zero changes in the minimization.

$$[y - F(x_i)]^T S_\epsilon^{-1} [y - F(x_i)]/n_y \le 1 \tag{6}$$

$$[x_i - x_{i-1}]^T S_x [x_i - x_{i-1}]/n_x \le 1 \tag{7}$$

When neither the inequality of Eq. 6 nor the inequality of Eq. 7 is reached after 15 iterations, the retrieval is considered as a failed retrieval.

## 3.2 Basic setting of the retrieval algorithm

In order to apply this theoretical framework to our retrieval algorithm, we define next the basic elements stated in the previous subsection.

The state vector $x$ contains the properties to be retrieved. In our case, they are the cloud optical thickness (COT) and the effective radius of water droplets ($R_{eff}$):

$$x = \begin{bmatrix} cot \\ R_{eff} \end{bmatrix} \tag{8}$$

It can be noted because the relationship between radiances and optical thickness has a logarithmic shape, that using *log(COT)* instead of COT in the state vector could have accelerated the convergence.

The a priori state vector was set to [10, 10µm] and the a priori covariance matrix $S_a$ was set to $10^8$. The latter was chosen very large to favor the measurements in the determination of the state vector (no a priori constraint).

The measurement vector $y$ contains the radiances ($R$) measured by OSIRIS at two wavelengths $\lambda_a$ and $\lambda_b$ for several view directions $\theta_i$ and is given in Eq. (9)

$$y = \begin{bmatrix} R_{\lambda_a}(\theta_1) \\ R_{\lambda_b}(\theta_1) \\ \vdots \\ R_{\lambda_a}(\theta_{n_\theta}) \\ R_{\lambda_b}(\theta_{n_\theta}) \end{bmatrix} \qquad (9)$$

The forward model is based on the adding-doubling method (De Haan et al., 1987; Van de Hulst, 1963) to solve the radiative transfer equation and simulate the radiances measured by OSIRIS for the corresponding observation geometries and wavelengths. It is a major element of the retrieval and describes the radiation interaction with the cloud, the surface, and the atmosphere while fixing several parameters (e.g. wind speed, cloud altitude…). We assume a standard atmosphere with a mid-latitude summer McClatchey profile (McClatchey et al., 1972) for the computation of

molecular scattering. As the two channels used in the retrieval are in atmospheric windows (as seen in Figure 1), the atmospheric absorption is not accounted for. It is not completely true, therefore the cloud optical thickness will be slightly underestimated and the effective radius slightly overestimated. Our case study is purely above an ocean surface. The reflection by the surface can affect the measured radiances even in cloudy conditions and particularly for optically thin clouds. The anisotropic surface reflectance of the ocean surface is characterized by a bidirectional polarization

distribution function (BPDF). We used the well-known Cox and Munk model to compute the specular reflection modulated by ocean waves (Cox and Munk, 1954) with a fixed ocean wind speed based on NCEP reanalysis of the National Oceanic and Atmospheric Administration (NOAA).

As in current operational algorithms, the cloud model used for the retrieval is a plane-parallel and homogeneous (PPH) cloud, which implies the independent pixel approximation (IPA). The case study is a liquid water cloud scene.

Therefore, we used a log-normal distribution for the size of particles, which are assumed spherical (Hansen and Travis, 1974) and described by an effective radius and an effective variance ($v_{eff}$). The altitude of the cloud is determined by the measurements of the LIDAR-LNG that was onboard the research aircraft Safire Falcon 20 during the airborne campaign. All simulations are monochromatic computations at the central wavelength of OSIRIS channels. The altitude of OSIRIS and the illumination and observation geometries are calculated based on the coordinates of the aircraft

inertial unit.

The Jacobian matrix $K$ includes the partial derivatives of the forward model to each element of the state vector (Eq. (10)). The columns of $K$ define then the sensitivity of the radiances (each with a specific wavelength - viewing angle configuration) to COT or $R_{eff}$. The rows of the Jacobian define the sensitivity of each radiance configuration to the two retrieved properties. The Jacobian Matrix is computed using finite differences.

$$K = \begin{bmatrix} \dfrac{\partial F_{\lambda_a}(\theta_1)}{\partial cot} & \dfrac{\partial F_{\lambda_a}(\theta_1)}{\partial R_{eff}} \\ \dfrac{\partial F_{\lambda_b}(\theta_1)}{\partial cot} & \dfrac{\partial F_{\lambda_b}(\theta_1)}{\partial R_{eff}} \\ \vdots & \vdots \\ \dfrac{\partial F_{\lambda_a}(\theta_{n_y})}{\partial cot} & \dfrac{\partial F_{\lambda_a}(\theta_{n_y})}{\partial R_{eff}} \\ \dfrac{\partial F_{\lambda_b}(\theta_{n_y})}{\partial cot} & \dfrac{\partial F_{\lambda_b}(\theta_{n_y})}{\partial R_{eff}} \end{bmatrix} \qquad (10)$$

### 3.3 Error characterization

During the retrieval process, every element is associated with a random or systematic error embedded in the error covariance matrix $S_\epsilon$. The account of errors in the inverse problem does not allow to have a unique value for the solution $x$ but instead a Gaussian probability distribution function (PDF) where $x$ is the expected value and $S_x$ is its covariance.

$S_x$ is calculated after a successful convergence with Eq. 4 using the Jacobian at the retrieved state and $S_\epsilon$. This posterior variance-covariance matrix can also be written as follow:

$$S_x = \begin{bmatrix} \sigma_{cot}^2 & 0 \\ 0 & \sigma_{r_{eff}}^2 \end{bmatrix} \qquad (11)$$

In this formulation, we have assumed that the two terms of the state vector are independent, thus the off-diagonal terms of $S_x$ are assumed to be zero. The use of Gaussian PDFs leads to computing the uncertainty on a particular parameter

$x_k$ as the square root of the corresponding diagonal elements of the covariance matrix $\sigma_k = \sqrt{S_{x_k}}$, where $k$ is the index of the parameter in the state vector $x$ (Eq. 11). We chose to express this uncertainty using the relative standard deviation (RSD) in % (Eq. 12). The RSD will be used to characterize the quality of the retrieval.

$$RSD = \left(\frac{\sigma_k}{x_k}\right) \times 100 \qquad (12)$$

$S_\epsilon$ represent the sum of the measurement ($S_{mes}$) and forward model variance-covariance matrix. Indeed, the forward model $F$ uses ancillary information provided by a set of fixed parameters $b$ (listed in section 3.3.2). Errors related to

an uncertain estimation of these fixed parameters are represented by the covariance matrix $S_{fp}$ described in the next section.

Besides the fixed parameters, the cloud model used in the radiative transfer computation can also be a source of uncertainty. The uncertainties of the retrieved parameters related to this approximation is regrouped in the covariance matrix $S_F$ described in the next section. $S_\epsilon$ is then addressed as a sum of these three components:

$$S_\epsilon = S_{mes} + S_{fp} + S_F \qquad (13)$$

Previous studies (Wang et al., 2016; Iwabuchi et al., 2016; Poulsen et al., 2012; Sourdeval et al., 2015) have already computed and presented the uncertainties of the retrieved cloud properties for all error contributions using $S_\epsilon$. Further, Walther and Heidinger (2012) use the optimal estimation framework to separate the contribution of measurement errors and several non-retrieved parameters. In our work, a similar framework was used to separate the contribution of each type of uncertainty including also the forward model uncertainties. The aim is to better quantify and understand the

limitation of using simplify forward model in such a cloud retrieval algorithm. It is realized by propagating the covariance matrices of errors from the measurement space into the retrieved state space (Rodgers, 2000). The gain matrix $G_y$, which represents the sensitivity of the retrieved quantities to the measurement, is then used:

$$G_y = S_x K^T S_\epsilon^{-1} \qquad (14)$$

The total variance-covariance matrix of the retrieved state vector ($S_x$) can then be decomposed into three contributions

(Eq. 15), with each term originating from its corresponding error covariance matrix.

$$S_x = S_{x_{mes}} + S_{x_{fp}} + S_{x_F} \qquad (15)$$

Each term in this equation is developed and discussed in the following three subsections.

### 3.3.1 Uncertainties related to the measurements

Any type of measurement is subject to errors. It is necessary to apply calibration processes to study the relationship
between the electrical signals measured by the detectors and the radiances and quantify its uncertainty. Calibration is done during laboratory experiments before the airborne campaign or the instrument launch into space (Hickey and Karoli, 1974). It can be done in situ if calibration sources are available onboard the sensor (Elsaesser and Kummerow, 2008) or can be vicarious (e.g. Hagolle et al. 1999) by using natural or artificial sites on the surface of the Earth. The uncertainties of the measurements remaining after the calibration processes are assumed, random and uncorrelated
between channels and can be consistently approximated by a Gaussian probability density function over the measurement space.

As errors between measurements are supposed to be independent, the covariance matrix of measurement noise ($S_{mes}$) is diagonal with dimensions equal to the measurements vector dimension ($n_y \times n_y$). The diagonal elements $\sigma_{mes_i}^2$ are the square of the standard deviation of the measurement errors. In our retrievals, we calculated the covariance matrix
based on 5% of measurement errors: $\sigma_{mes} = R_{\lambda,\theta} \times 5\%$.

$$S_{mes} = \begin{bmatrix} \sigma_{mes_1}^2 & 0 & \dots & 0 \\ 0 & \sigma_{mes_2}^2 & \dots & 0 \\ \vdots & \vdots & \ddots & \vdots \\ 0 & 0 & \dots & \sigma_{mes_{n_y}}^2 \end{bmatrix} \tag{16}$$

The error covariance matrix for the retrieved parameters due to measurement errors is then expressed by mapping the covariance matrix $S_{mes}$ from the measurement space to the state space by using the gain matrix $G_y$:

$$S_{x_{mes}} = G_y S_{mes} G_y^T \tag{17}$$

The uncertainty on a particular parameter $x_k$ originating from the measurement errors is defined as the square root of the corresponding diagonal element corresponding to the standard deviation $\sigma_{k_{mes}} = \sqrt{S_{x_{mes\,k}}}$. It is expressed using the RSD (mes) as in Eq. (12).

### 3.3.2 Uncertainties related to the fixed parameters

Any retrievals from remote sensing observations require prior knowledge of several unknown parameters used in the
forward model computation. Those parameters are not retrieved due to a lack of sufficient information. To compute the fixed parameters (fp) errors, we quantified the possible error in our estimation of the fixed model parameters. In our case study, these parameters are the altitude of the cloud (alt), the effective variance of the cloud droplet size distribution ($v_{eff}$), and the ocean wind speed (ws). These errors are considered to be independent and random under the assumption of linearity of the radiances around the fixed parameters. They are set in the diagonal covariance
matrix $S_{\sigma fp}$. They are weighed by $K_{fp}$ the Jacobian matrix containing the gradient of the forward model with respect to

the fixed parameters. Finally, as previously, the errors are mapped from the measurements space to the state vector space through $G_y$ to estimate their contribution to the retrieval uncertainty as follows:

$$S_{x_{fp}} = G_y S_{fp} G_y^T = G_y K_{fp} S_{\sigma fp} K_{fp}^T G_y^T \tag{18}$$

Each column in $K_{fp}$ and $S_{\sigma fp}$ is dedicated to one fixed parameter. Therefore, we can separate the contributions of every
element of the fixed parameters vector as follows:

$$S_{x_{fp}} = S_{x_{fp,alt}} + S_{x_{fp,ws}} + S_{x_{fp,v_{eff}}}$$

(19)

Each covariance matrix from the right side of Eq. (19) is developed as shown in Eq. 20. $\sigma_{b_i}$ is the standard deviation of the fixed parameter error and $K_{b_i}$ is a column vector containing the gradient of the forward model in regard of the
same fixed parameter $b_i$.

$$S_{x_{fp,b_i}} = G_y K_{b_i} \sigma_{b_i}^2 K_{b_i}^T G_y^T \tag{20}$$

In order to develop $S_{x_{fp,b_i}}$ for each element of $b$, the forward model has been constructed in a flexible way that permits to initiate small variations of any fixed parameter and then calculate the partial derivatives of the forward model in regard to the ancillary data, called the Jacobians of the fixed parameters $K_{alt}, K_{v_{eff}}$, and $K_{ws}$.

The last elements needed to resolve Eq. (20) are the errors or standard deviations of the cloud altitude, the effective variance of water droplets and the ocean wind speed, $\sigma_{alt}$, $\sigma_{v_{eff}}$ and $\sigma_{ws}$ respectively.

The values and the uncertainties of these fixed parameters are chosen according to the experimental setup of the campaign. To estimate the uncertainties originating from the fixed cloud altitude, we used the opportunity of having the LIDAR-LNG aboard the aircraft, which gives the backscattering signal obtained around the case study of
CALIOSIRIS. From 11:01:06 to 11:03:06 (the time when the same cloud scene is apparent), the cloud altitude varies between 5.57 and 5.73 km in our cloud scene. For practical reasons related to the radiative transfer code, we use a value of 6km for the cloud top altitude and a standard deviation of $\sigma_{alt} = 0.16$ km (3% of the cloud altitude). This value is low thanks to the knowledge provided by the Lidar.

Concerning the effective variance $v_{eff}$, to which the polarized radiance is highly sensitive in the supernumerary arcs
near the cloud bow (Bréon and Goloub, 1998), we fixed a value of 0.02 based on the number of supernumerary bows in the polarized radiances (not shown). After simulating radiances with several values of $v_{eff}$, we choose to add a $\sigma_{v_{eff}} = 0.003(15\%)$ possible error in the estimation of this parameter. As the value of $v_{eff}$ was fixed using the polarization measurements of OSIRIS, this uncertainty is weak and not representative of all situations.

For the ocean wind speed fixed to 8 m/s obtained from the database of the National Oceanic and Atmospheric
Administration, we used an error $\sigma_{ws} = 0.8$ m/s (10%). It covers the possible sources of error in surface wind speed retrievals.

### 3.3.3 Uncertainties related to the forward model

Forward models are usually formulated around some limitations and assumptions that can contribute to the uncertainty of the retrieved parameters. The forward model used to simulate the radiances measured by OSIRIS follows the cloud

plane parallel assumption. This assumption is known to cause errors on the retrieved parameters (see section 1) that can be assessed and included in the total uncertainty. The evaluation of these modeling errors requires an alternative forward model $F'$ that includes more realistic physics. The contribution of this error is represented by the following equation:

$$S_{x_F} = G_y S_F G_y^T \tag{21}$$

$S_F$ is diagonal with dimensions equal to the measurement vector dimensions ($n_y \times n_y$). Each diagonal element is the square of the difference between radiance computed for a specific direction with the simplified forward model $F$ and the one computed with the more realistic forward model $F'$ while maintaining the same state vector and the same fixed parameter vector $b$: $\left(F(x,b) - F'(x,b)\right).\left(F(x,b) - F'(x,b)\right)^T$.

The simplified model used for the retrieval can lead to biased retrieved parameters. In this case, the bias due to the
model will be included in the Gaussian PDF width, resulting in an overestimation of the uncertainties.

The uncertainties related to the cloud vertical homogeneity and the cloud horizontal homogeneity are quantified separately. In the following, we present the elements of the forward model used to quantify the uncertainties of these assumptions.

*Non-uniform cloud vertical profile model*

The vertically heterogeneous cloud model to assess the uncertainties of the assumed homogeneous cloud model is described by:

- an effective radius profile and possibly an effective variance profile but for simplification, we will consider that $v_{\text{eff}}$ is constant over the entire vertical profile with a value of 0.02.

- an extinction coefficient ($\sigma_{ext}$) profile

- a cloud geometrical thickness (CGT) characterized by the difference between the altitude of the cloud top ($z_{top}$) and the cloud base ($z_{bot}$). The values of CGT, $z_{top}$ and $z_{bot}$ are fixed based on the LIDAR measurements.

The effective radius and extinction coefficient profiles are computed using an analytical model already introduced in (Merlin, 2016). It is based on adiabatic cloud profiles, which are described and used in several studies (Chang, 2002;
Kokhanovsky and Rozanov, 2012). In the adiabatic scheme, the effective radius increases with altitude. However, several studies proved that a simple adiabatic profile is not sufficient to describe a realistic cloud profile (Platnick, 2000; Seethala and Horváth, 2010; Nakajima et al., 2010; Miller et al., 2016). Depending on the maturity of the cloud, turbulent and evaporation processes can reduce the size of droplets at the top of the cloud and/or collision and coalescence processes can increase the size of the droplets in the lower part of the clouds as observed by Doppler Radar
(Kollias et al., 2011). The profile used in this study aims to represent the case of droplet size reduction at the top of the cloud, but other and more sophisticated and representative profiles can be used (Saito et al., 2019).

The description of this more realistic vertical cloud profile is obtained with two adiabatic profiles (Figure 3) that are joined at the altitude of maximum LWC called $z_{max}$:

- The first profile from $z_{bot}$ to $z_{max}$ is considered adiabatic.

- The second profile from $z_{max}$ to $z_{top}$ follows an adiabatic LWC profile decreasing with altitude.

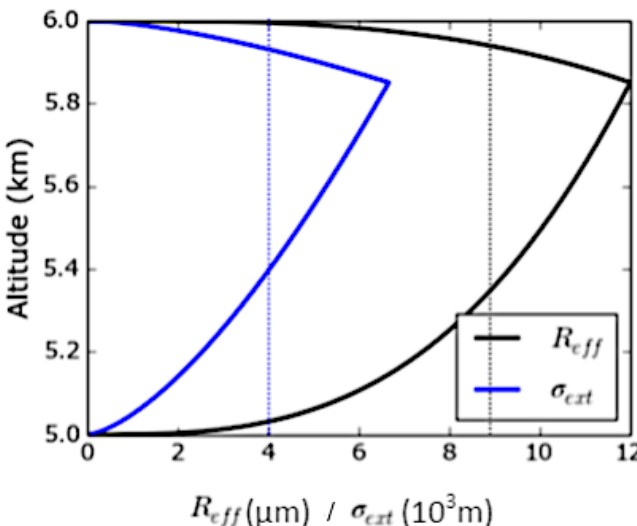

**Figure 3: The heterogeneous vertical profile of effective radius (black line) and extinction coefficient (blue line) used to assess uncertainties due to the assumption used for the vertical profile. The equivalent homogeneous vertical profiles are shown in dashed lines. The cloud is between 5 and 6 km. The maximum extinction coefficient and effective radius are 6.6 km⁻¹ and 12 μm respectively and the altitude z_max is 5.85km**

Considering that LWC is equal to zero at the base and top of the cloud, and relying on the linear variation model of the LWC with altitude ($z$) established in (Platnick, 2000), we can write that:

$$LWC(z) = LWC_{max} \frac{z - z_{bot}}{z_{max} - z_{bot}}; z \in [z_{bot}, z_{max}]$$

$$LWC(z) = LWC_{max} \frac{z_{top} - z}{z_{top} - z_{max}}; z \in [z_{max}, z_{top}]$$

(22)

The profiles of effective radius (Eq. 23) and extinction coefficient (Eq. 24) can then be computed by considering that

the particle concentration is constant over the entire cloud which makes it possible to obtain analytical functions of LWC, $R_{eff}$ and $\sigma_{ext}$.

$$R_{eff}(z) = R_{eff_{max}} \left( \frac{z - z_{bot}}{z_{max} - z_{bot}} \right)^{\frac{1}{3}}; z \in [z_{bot}, z_{max}]$$

$$R_{eff}(z) = R_{eff_{max}} \left( \frac{z_{top} - z}{z_{top} - z_{max}} \right)^{\frac{1}{3}}; z \in [z_{max}, z_{top}]$$

(23)

$$\sigma_{ext}(z) = \sigma_{ext_{max}} \left( \frac{z - z_{bot}}{z_{max} - z_{bot}} \right)^{\frac{2}{3}}; z \in [z_{bot}, z_{max}]$$

(24)

$$\sigma_{ext}(z) = \sigma_{ext_{max}} \left( \frac{z_{top} - z}{z_{top} - z_{max}} \right)^{\frac{2}{3}}; z \in \left[ z_{max}, z_{top} \right]$$

A form factor $p$ (Eq. 25) allows the adjustment of the altitude $z_{max}$ where the extinction coefficient and the effective radius are the largest:

$$p = \frac{z_{top} - z_{max}}{z_{top} - z_{bot}} \tag{25}$$

This unitless parameter $p$ varies from 0 to 1 representing the shape of the profile. The value 0 corresponds to $z_{max} = z_{top}$ (adiabatic cloud) and the value 1 corresponds to $z_{max} = z_{bot}$ (a reverse adiabatic cloud with a negative gradient of water content). In the following results, a value of 0.15 is assigned to this parameter which allows having a profile close to the one studied in (Miller et al., 2016) from large-eddy simulations (LES) cloud scenes.

To assess the error due to the vertical heterogeneity of the cloud, we need to specify the maximum value of the extinction coefficient $\sigma_{ext_{max}}$ and the effective radius $R_{eff_{max}}$ of the vertically heterogeneous cloud, corresponding to the "equivalent" homogeneous clouds. Several options are possible for these values. We choose to use Eq. (26) to assign $\sigma_{ext_{max}}$ which leads to the same integrated extinction profile and Eq. (27) to assign $R_{eff_{max}}$ to ensure that the mean $R_{eff}$ of the heterogeneous vertical profile is equal to the $R_{eff}$ of the homogeneous cloud ($R_{eff_F}$). $\sigma_{ext_{max}}$ and $R_{eff_{max}}$ are found analytically by integrating the profiles described in Eq. (23) and Eq. (24).

$$\sigma_{ext_{max}} = \frac{5}{3} CO\,T_F / \left( z_{top} - z_{bot} \right)$$
(26)

$$R_{eff_{max}} = \frac{4}{3} R_{eff_F}$$
(27)

A vertically heterogeneous cloud is computed for each pixel using the retrieved value based on the homogeneous assumption. The error-covariance matrix describing the error due to the simple homogenous cloud assumption (Eq. 21) is calculated from the difference between radiances computed with homogeneous and heterogeneous vertical profiles, denoted $F$ and $F'$ respectively.

*The 3D radiative transfer model*

The other assumption that might affect the retrieved cloud optical properties in the current operational algorithms is the horizontally plane-parallel and homogeneous (PPH) assumption for each observed pixel. It implies that each pixel is horizontally homogeneous and independent of the neighboring pixels known as the independent pixel approximation (IPA). The homogeneous PPH assumption affects the cloud-top radiances and leads to differences between 1D and 3D radiances that are the result of several effects discussed in numerous publications and briefly summarized in section 1. This PPH assumption includes errors known as the PPH bias due to the subpixel variations of the cloud and errors related to the photon horizontal transport between columns (IPA error). At the high spatial resolution of OSIRIS (less than 50 m), it was shown from airborne data that the dominating effect is related to the IPA error (Zinner et Mayer,

2006). In the following, we consider thus only this error and assume that the pixel is homogeneous at the measurement scale.

To assess the uncertainties in the retrievals arising from this assumption, Eq. (21) is used. $S_F$ is then the difference between the radiances computed with a 1D radiative transfer code (1D-RT), following the Adding-doubling method (Hansen and Travis, 1974), and the radiances computed with a 3D radiative transfer (3D-RT) code called 3DMCPOL (Cornet et al., 2010). The 3D simulations use, for each pixel, the COT and $R_{eff}$ retrieved using the PPH assumption. Errors on cloud model assumptions are assessed independently so a vertical homogeneous profile is assumed. We also

assume a flat cloud top, which leads to underestimated differences and errors as cloud top variation may increase the differences between 3D and 1D radiances (Várnai and Davies, 1999; Várnai, 2000). The differences are thus mainly due to the lateral photon transport which tends to smooth the radiances fields compared to their 1D counterpart (Davis et al. 1997) and to the cloud heterogeneity along the line of sight (e.g. Fauchez et al., 2018).

## 4 Retrieval and uncertainty estimation for a liquid cloud case study measured by OSIRIS

Our strategy to assess the different types of uncertainty follows two steps. In a first step, we retrieve COT and $R_{eff}$ using a bispectral multi-angular method by considering the uncertainties related to the measurement errors alone. We use a weakly absorbing channel centered at 1240 nm that is mainly sensitive to COT, and a partially absorbing channel centered at 2200 nm, and thus sensitive to $R_{eff}$. In this case study, up to 13 viewing angles are available for each pixel. In the first step, only the measurement errors are accounted for and included in $S_\epsilon$. This error is usually well

characterized and does not change once the measurements are realized. Not considering the other errors at this stage allows benefiting from a faster retrieval algorithm without the calculation of $K_b$ and the heavy computation cost of heterogeneous cloud profiles and 3D-RT calculations. The second step consists of computing the errors due the non-retrieved parameters and due to the assumption of a vertically and horizontally homogeneous cloud for the retrieval of $R_{eff}$ and COT.

It should be noted that the parameters retrieved in the first step may be biased, in particular due to the use of a simplified cloud model to connect the state vector to the measurements. We assume that the estimation of the uncertainties performed in the second step is however correct if the variations predicted by the simplified and the realistic models around the retrieved values (potentially biased) and around the true values are identical. This is correct with a linear forward model but can be a too strong assumption in cloud retrieval regarding the non-linearity of the relationship of

the radiances in function of cloud parameters. A way to test this assumption would be to use numerical experiments.

In Figure 4, COT (a) and $R_{eff}$ (b) retrieved from multi-angular SWIR radiances are presented. Spatial variations are mainly due to variations in the observed cloud structures. The COT range is between 0.5 and 6 with a mean value of 2.1. Some values of COT are very small, but no clear sky pixel is present. $R_{eff}$ varies between 2 and 24 µm around a mean value of 8.8 µm. Figure 4c presents the normalized cost function, which is less or equal to one when the retrieval

successfully converges according to Eq. 6 (convergence of Type 1). In the case of multi-angular measurements, the normalized cost function is often above one meaning that the simulated radiances do not fit the measurements while considering the measurements error covariance only. This comes from the attempt to fit the measured radiances from

all the available viewing directions with a simple forward model, far from reality. The retrieval stops thus mainly according to Eq. 7 (convergence of Type 2) indicating that the state vector remains almost constant between two successive iterations. When neither Eq. 6 nor Eq. 7 are achieved the retrieval fails. For the whole scene, failed retrievals account for 3.3% of the pixels. The failure may be associated with pairs of radiances outside the LUT that can occur for several reasons well documented in Cho et al. (2015).

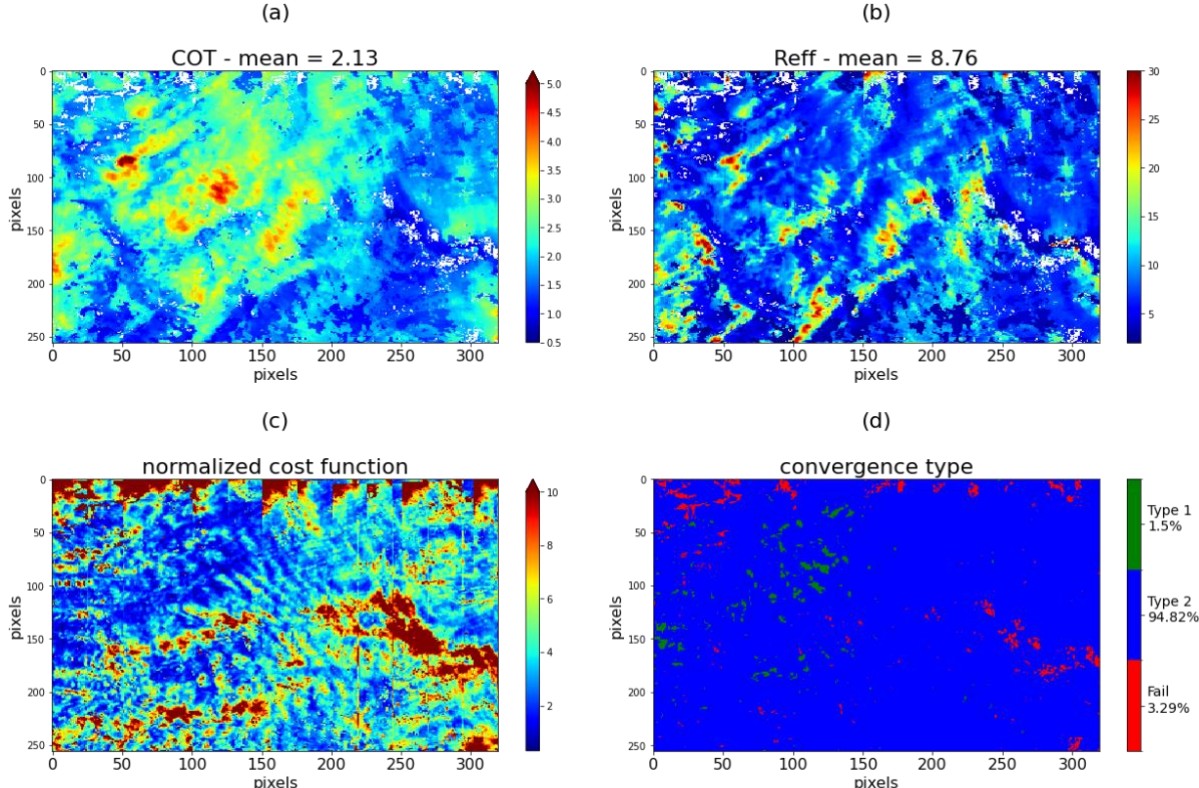

**Figure 4: COT (a) and $R_{eff}$ (b) retrieved using a multi-angular bispectral method from a liquid cloud case observed during the CALIOSIRIS airborne campaign on 24 October 2014 at 11:02 (local time).** Pixels associated to failed retrievals are represented by white pixels. (c) Normalized cost function. (d) Convergence type (Eq. 6 for Type 1 and Eq. 7 for Type 2) and failed retrieval.

As detailed in section 3.3, the final error is divided into three categories. Figure 5 shows the uncertainties originating from a 5% measurement error on the retrieved COT, RSD COT (mes), and on the retrieved $R_{eff}$, RSD $R_{eff}$ (mes). RSD COT (mes) ranges from 0.5 to 5% with a mean value of 3.2% while RSD $R_{eff}$ (mes) ranges from 2 to 12% with a mean value equal to 6.3%. These uncertainties are plotted according to their respective values in panels (c) and (d). RSD COT (mes) increases with the magnitude of the retrieved COT, as tends to do RSD $R_{eff}$ (mes) with $R_{eff}$ for values until 12 µm. The uncertainties due to measurement errors are low, especially for optical thickness (less than 5%). This is related to the quasi-linearity and the steep slope of the radiance as a function of COT in this cloud regime (small COT). When the radiance-COT relationship is quasi-linear, the sensitivity of the forward model to COT is high, which consequently lead to parameters retrieved with a high accuracy (low RSD). When COT increases, the gradient of the radiance-COT relationship decreases causing larger uncertainties.

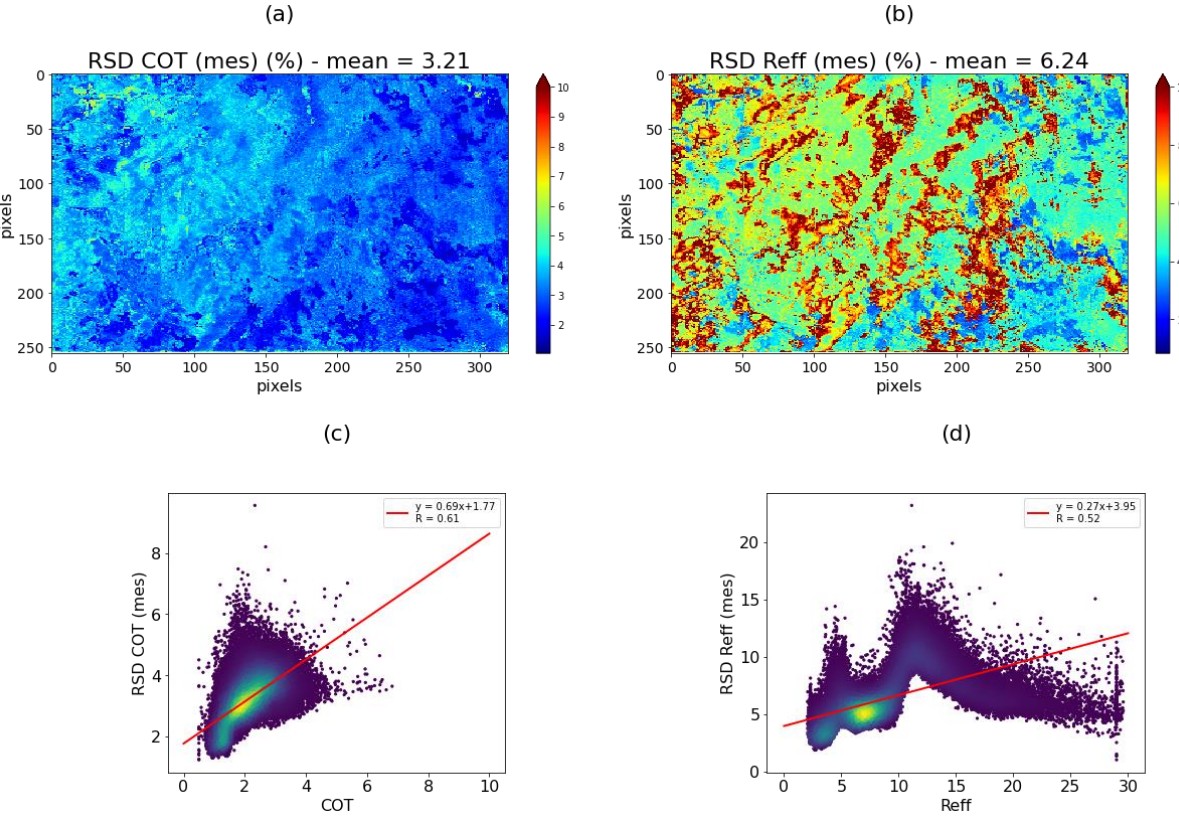

**Figure 5: Uncertainties (RSD) in % of COT (a) and $R_{\text{eff}}$ (b) originating from the measurement errors for the case study of CALIOSIRIS. COT uncertainties as a function of COT (c). $R_{eff}$ uncertainties as a function of $R_{eff}$ (d).**

The second type of uncertainty is related to the fixed parameters in the forward model. In Figure 6, we show the uncertainty on COT and $R_{\text{eff}}$ in % due to an incorrect estimation of each fixed parameter in the forward model. Panels (a) and (b) represent the uncertainties originating from the fixed cloud altitude, RSD COT (alt) and RSD $R_{eff}$ (alt) respectively. Both show very small uncertainties with values close to zero. In fact, in the visible range, the altitude of the cloud mainly determined the amount of Rayleigh scattering that occurs above the cloud. This type of scattering is

dominant at shorter and visible wavelengths and becomes negligible at the studied wavelengths (1240 and 2200 nm). Consequently, at these wavelengths, an error in the fixed cloud altitude does not contribute to the uncertainty on the retrieved COT and $R_{\text{eff}}$.

The (c) and (d) panels in Figure 6 represent the uncertainties of the retrieved COT and $R_{\text{eff}}$ originating from the fixed effective variance of the particle size distribution. They are nearly null on COT with a mean value of 0.05% as the 15%

uncertainty on the value of $v_{\text{eff}}$ (0.02) does not modify the total radiances. On the other hand, RSD $R_{\text{eff}}$ ($v_{eff}$) reaches 0.5% with a mean value of 0.25%. Indeed, $v_{\text{eff}}$ modifies the width of the cloud droplet distribution and consequently slightly the absorption by cloud droplets, resulting in a larger error. For $R_{\text{eff}}$ higher than 15 µm, the relationship between SWIR radiances and $R_{\text{eff}}$ tends to flatten which make them less sensitive to $v_{\text{eff}}$ and thus the uncertainties are smaller than 0.1%. We remind that we fixed the value of $v_{\text{eff}}$ using multi-angular polarized measurements of OSIRIS, which

leads to choose a weak uncertainty for $v_{eff}$ (15%). In case of lack of information on $v_{eff}$ in the measurements, the uncertainty should be higher and thus the errors due to the non-retrieved effective variance. Platnick et al. (2017) obtain 2% and 4% uncertainty for COT and $R_{eff}$ respectively for $v_{eff}$ ranging between 0.05 and 0.2.

Panels (e) and (f) in Figure 6 show that an error in the estimation of the ocean wind speed affects the retrieved COT and $R_{eff}$ mainly for small COT. The water-air interface is reflecting mainly in the specular direction, but the ocean

being not perfectly smooth, the bright surface (named glitter) is enlarged by the waves formed by the wind. The higher the surface wind speed is, the greater the amplitude of the waves is, leading to a larger reflection angle (wider sun-glint). The Sun-glint reflection is seen by OSIRIS only for very small values of COT and implies uncertainties of the retrieved parameters of about 0.5%. In the case of broken clouds, the errors resulting from the ocean wind speed uncertainties would be larger. At higher COT, the surface is non-apparent to OSIRIS measurements, and uncertainties

are thus close to zero.

We note that all the uncertainties of the studied fixed parameters remain below 1%, which shows that retrieval of all the COT-$R_{eff}$ couples does not have a high dependence on the fixed forward model parameters.

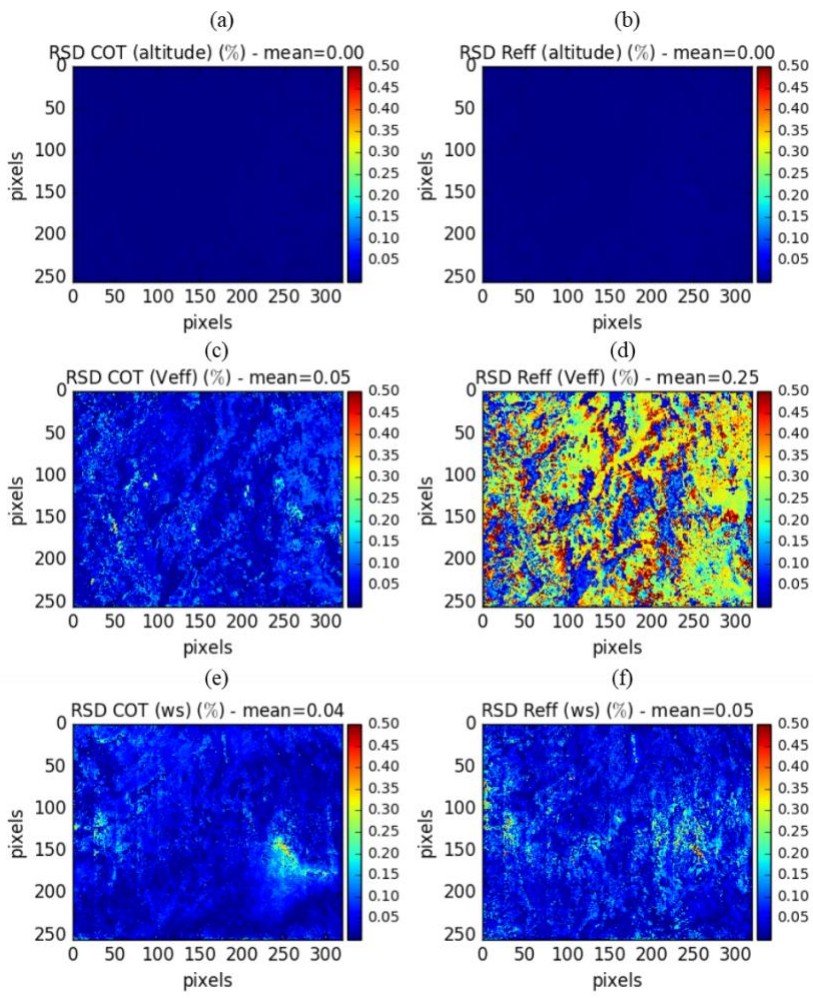

**Figure 6: The uncertainties RSD (%) of COT (left column) and $R_{eff}$ (right column) originating from the non-retrieved parameter errors: altitude (a and b), the effective variance of water droplet distribution (c and d), and the surface wind speed (e and f).**

The uncertainties due to the assumptions of the forward model are presented in Figure 7. The panels (a) and (b) represent the uncertainties of COT and $R_{eff}$ respectively, originating from the vertically homogeneous assumption. RSD COT (Fpv) ranges between 1 and 8% with a mean value of 4.9% while RSD $R_{eff}$ (Fpv) varies from 2 to 20% with a mean value of 13.3%. We note that when the cloud is optically thin (left part of the image), RSD COT (Fpv) and RSD $R_{eff}$ (Fpv) tend to be lower. When the extinction is small, the radiations penetrate deeper into the cloud and bring information on the whole cloud, similar to the one obtained with the homogenous vertical profile. The differences between radiances coming from the vertical heterogeneous and homogeneous profiles are thus small since the integrated extinction over the cloud is approximately the same in both cases. For larger COT, the radiations penetrate less in the cloud and are only affected by the upper part of the cloud where the extinction coefficient is different from one profile to another. In this case, RSD COT (Fpv) and RSD $R_{eff}$ (Fpv) are larger up to 8% and 20% respectively.

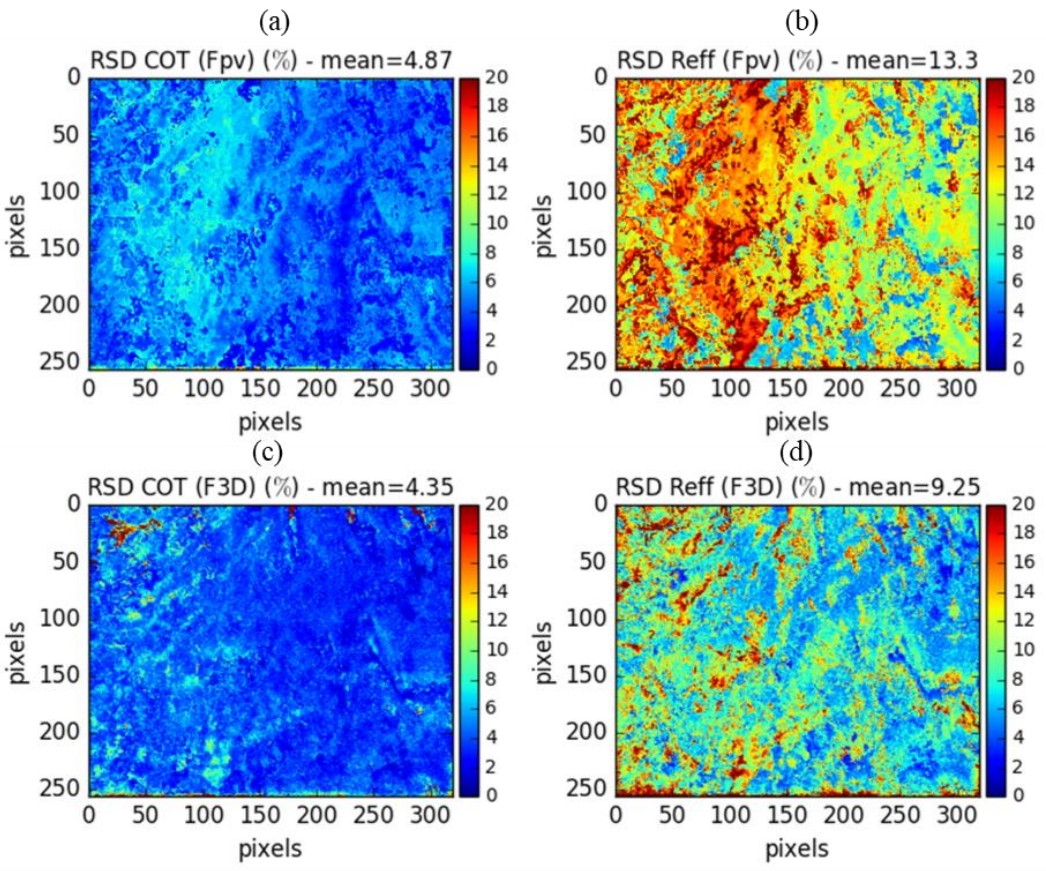

**Figure 7: The uncertainties (%) on COT and $R_{eff}$ originating from the assumptions in the forward model when not considering the heterogeneous vertical profile (a and b) and the 3D radiative transfer (c and d).**

The uncertainties originating from the use of a 1D radiative transfer code instead of a more realistic 3D radiative transfer are represented in Figure 7 (c) and (d) for COT and $R_{eff}$ respectively. RSD COT (F3D) ranges between 1 and 20% with a mean value of 4.35%, while RSD $R_{eff}$ (F3D) varies from 2 to 18% with a mean value of 9.25%. We remind here that, given the high spatial resolution of OSIRIS measurements, we consider the PPH bias as negligible and do not account for the sub-pixel variability of cloud properties in the 3D radiative transfer simulation.

Considering the solar zenith incidence angle (59º), illumination and shadowing effects can also be present depending on the viewing geometries and roughness of the radiative fields (Marshak et al. 1995b, Varnai, 2000)  However, in this work, we are dealing with flat cloud tops that induce weaker 3D effects than bumpy cloud tops (Varnai et Davies, 1999). In addition, with multi-angular measurements, the same cloudy pixel is viewed under different viewing angles, which may tend to mitigate the influence of illumination and shadowing effects.

At this scale, the effects related to the Independent Pixel Approximation (IPA) (Oreopoulos and Davies, 1998) are dominant since the horizontal transfers of photons between pixels are important. The smaller the column horizontal sizes are considered, the more the real behavior of radiation in the atmosphere will be misrepresented. The horizontal radiation transport (HRT) tends to smooth the radiative field by increasing or decreasing the radiances according to the optical thickness gradient between the considered pixel and its neighbors. This effect is shown in Figure 8. The panels (b) and (d) representing the reflectances computed with 3DMCPOL at 1240 and 2200 nm respectively show a smoothest field compared to the reflectances computed with a 1D radiative transfer model in panels (a) and (c). The variabilities in the 3D radiative field are indeed less pronounced compared to the 1D field.

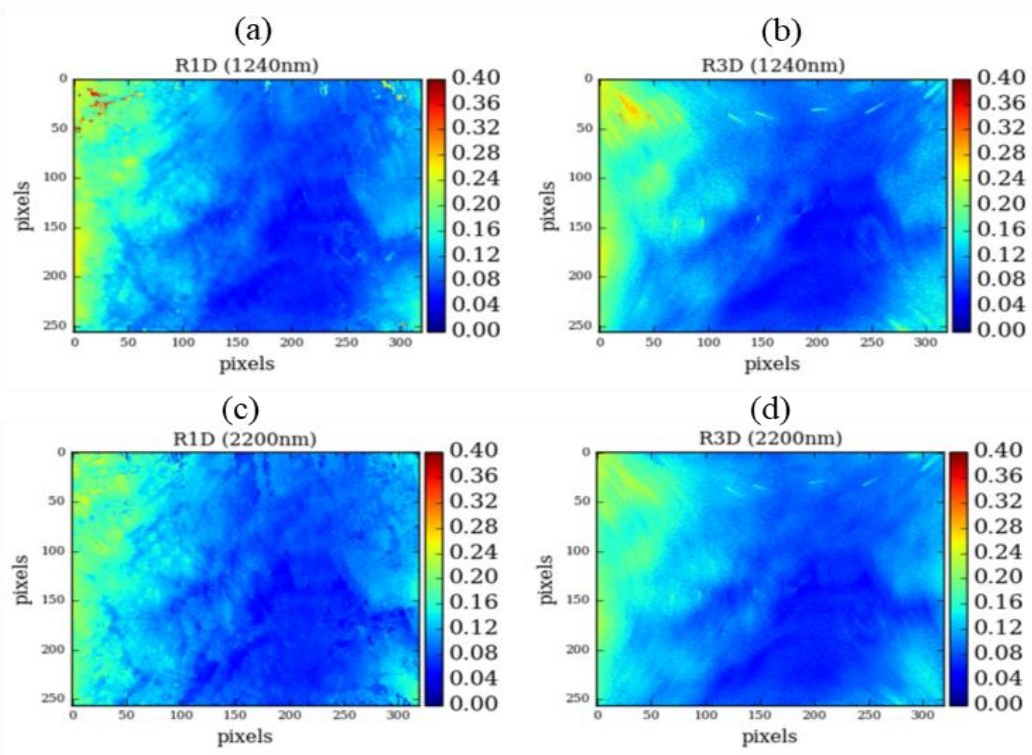

**Figure 8: The simulated 1D (a) and 3D (b) reflectances at 1240 nm using the retrieved COT and $R_{\text{eff}}$ presented in Figure 4 for the central image. (c) and (d) are the same as (a) and (b) but for 2200 nm.**

In Figure 9, the histograms of the relative difference between the radiances computed in 1D (R1D) and the radiances computed in 3D (R3D) at 1240 nm for different bins of optical thickness are plotted. We can see the shift of the histograms from negative values for small optical thickness (R1D < R3D) towards positive differences for larger optical thickness (R1D > R3D) that is explained by the horizontal radiation transport between columns.

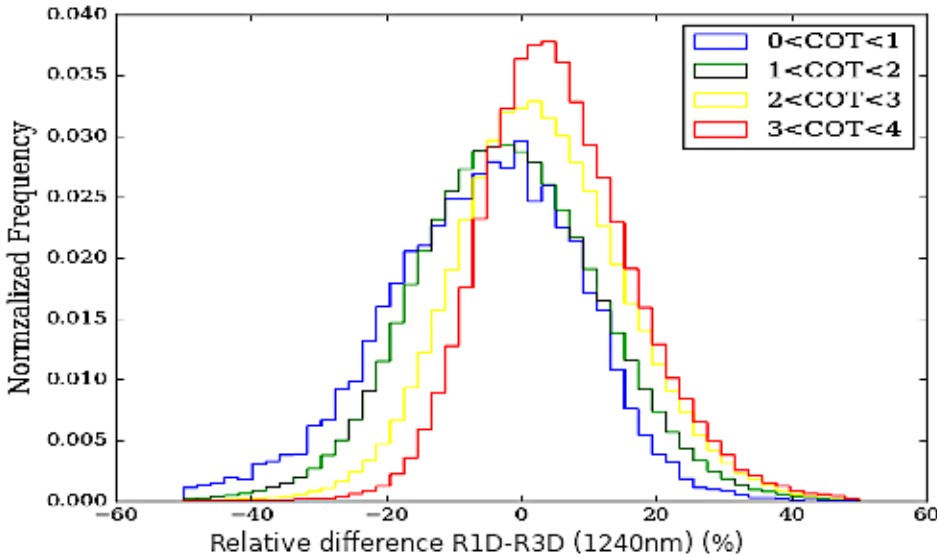

**Figure 9: Histograms of the relative difference between the reflectances computed in 1D and 3D at 1240 nm for the central image. Each histogram corresponds to a domain of COT.**


Overall, we note that the uncertainties due to the forward model assumption are much more important than the ones due to the fixed parameters. The retrieval is not sensitive to small variations in the fixed parameters. However, while assessing uncertainties due to the vertical profile or radiative transfer assumption, we change the parameters that our forward model is proven to depend on, thus changes in the integrated profile can lead to relatively large variations in

the radiance fields, and consequently large uncertainties.

## 5 Advantages of using multi-angular versus mono-angular information

The same strategy applied in section 4, is applied using the bispectral mono-angular method used for the MODIS instrument for example. For the mono-angular bispectral approach, the measurement vector $y$ for each pixel contains two mono-angular total radiances, one at 1240 nm and the other at 2200 nm. The mono-angular direction corresponds

to that of the central image of the multi-angular sequence used to retrieve COT and $R_{\text{eff}}$ with the multi-angular measurements.

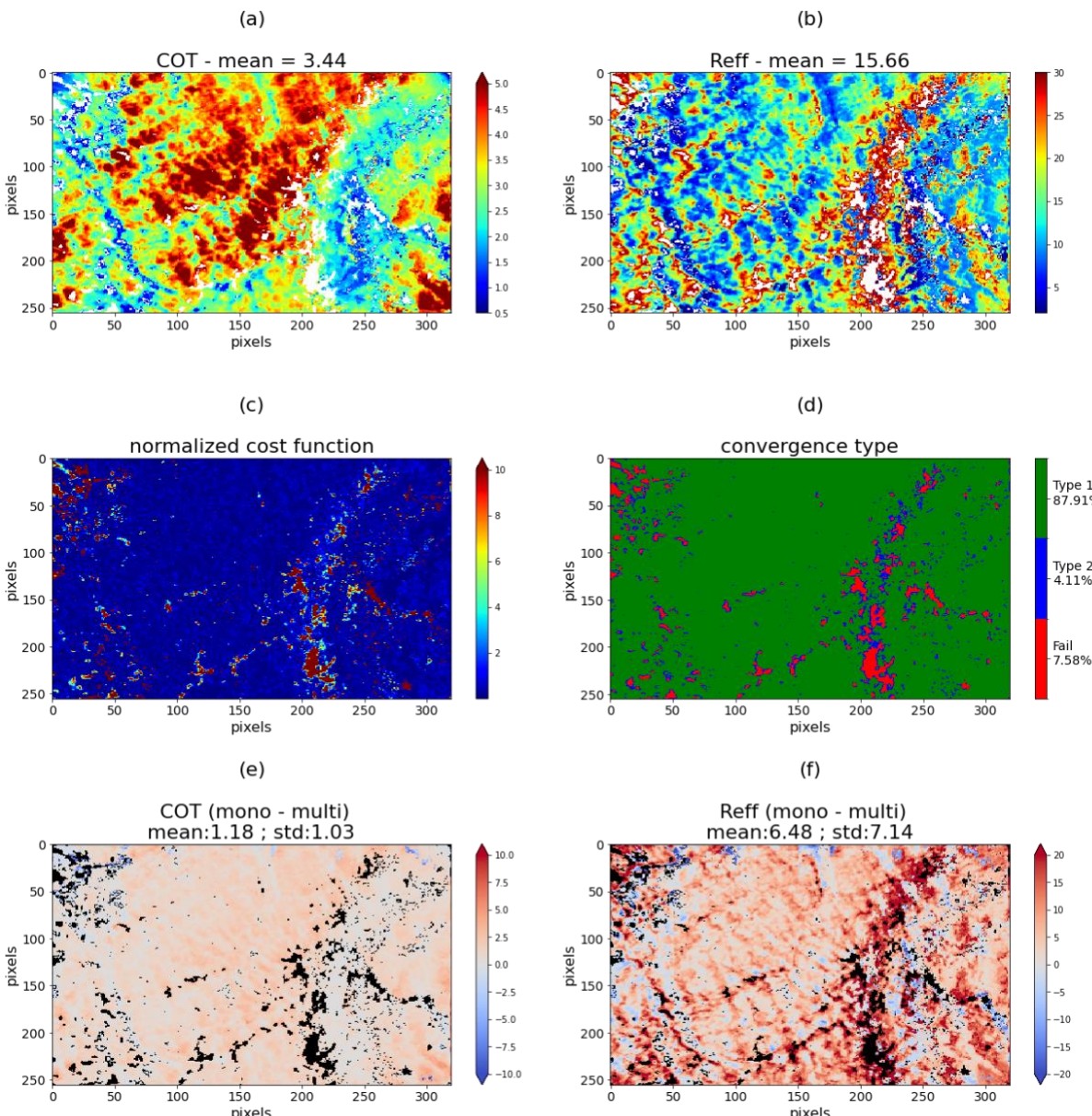

**Figure 10: COT (a) and $R_{\text{eff}}$ (b) retrieved using the mono-angular bispectral method for the CALIOSIRIS liquid cloud case study on 30 June 2014 at 11:02 (local time). Pixels associated with failed retrievals are represented by white pixels. (c) Normalized cost function. (d) Convergence type (Eq. 6 for Type 1 and Eq. 7 for Type 2) and failed retrieval. Differences between mono-angular and multi-angular retrieval for retrieved optical thickness (e) and retrieved effective radius (f).**

The results are presented in Figure 10. The retrieved COT over the whole field varies between 1 and 12 with a mean value equal to 3.44. Compared to multi-angular measurements (mean COT of 2.13), the retrieved COT values tend to be higher. The range of retrieved $R_{\text{eff}}$ has a mean value of 15.65 µm, compared to 8.76 µm for multi-angular retrieval. Mono-angular retrieval is particularly affected by the high value of $R_{\text{eff}}$ retrieved around the scattering angles 130-140° where the sensitivity of 2200 nm radiances to the water droplet size is known to be small. This area corresponds also to a more important number of failed retrievals. As a matter of fact, (Cho et al., 2015) have indeed shown that in liquid marine cloud cases, the phase functions of different $R_{\text{eff}}$ converge to the same value for these scattering angles ranges leading to the failure of water droplet size retrieval from MODIS measurements. This reduced sensitivity also explains the high uncertainty on $R_{\text{eff}}$ due to measurement errors around the cloud bow (Figure 11). The

smaller sensitivity on $R_{eff}$, in this case, is not limited to the cloud bow directions and supernumerary bows but is also visible at some regions of small scattering angles (70-80°) that can be affected by specular reflection over the ocean.

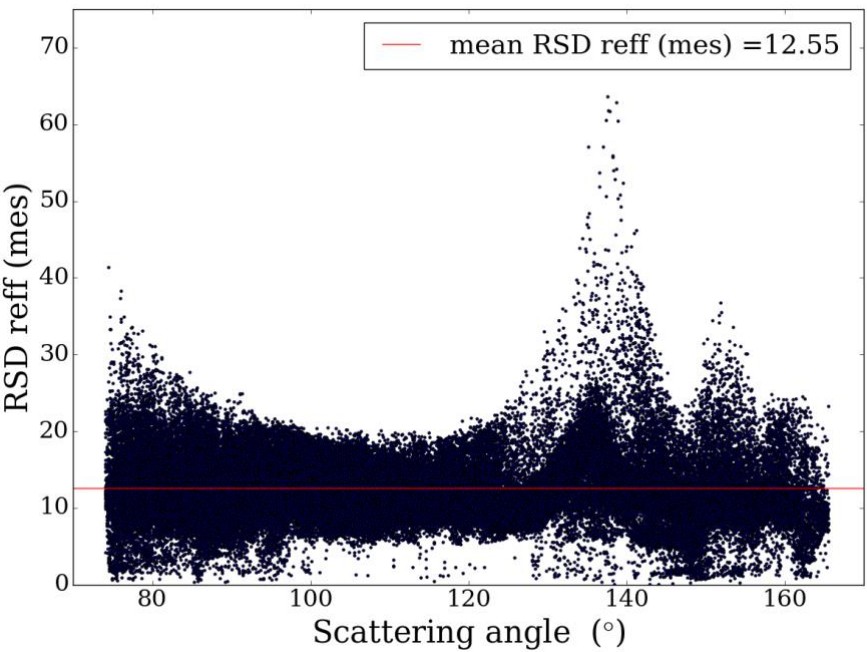


**Figure 11: Uncertainties on the effective radius originating from the measurement errors, *RSD $R_{eff}$ (mes)* as a function of the scattering angle for the mono-angular retrieval. The red line represents the mean *RSD $R_{eff}$ (mes)* = 12.55%.**

Multi-angular retrieval presents the major advantage, that no aberrant values of $R_{eff}$ are retrieved near the scattering angles at 140° (comparing Figure 4b to **Figure** 10b). The multi-angular measurements contain more information and

allow to resolve the problem encountered with the mono-angular bispectral method, which is also clear in the reduction of the failed convergences from 7.6% to 3.3%. In the overall scene, smaller $R_{eff}$ values are obtained. A smallest effective radius leads to an increase in the backward scattering and so in the reflected radiance, which results in a lower retrieved optical thickness.

Excepted in case of failed retrievals that occur for values outside the LUT ranges, the relation between radiances and

COT-$R_{eff}$ being monotonical, the mono-directional method allows to always find retrieved values, that is a pair of COT and $R_{eff}$ that matches the measured radiances. However, these values can be more or less far from the real values. A normalized cost function value (Figure 10c) less or equal to one is thus not necessarily an indication of an accurate retrieval, but only that a fit occurred. On the other hand, multi-angular retrieval increases the constraint on the forward model, which makes it much more challenging to find a solution allowing to fit the measurements. The retrieved state

is then consistent at the best with all the measurements associated with different viewing angles.

To compare the uncertainties of the two retrievals, we use the relative standard deviation (RSD) to be consistent with the previous results. In **Figure** 12, we present the spatial average of the different types of errors, presented in section 4, for the mono-angular method (light green for COT and light blue for $R_{eff}$) in comparison with the multi-angular method (dark green for COT and dark blue for $R_{eff}$). We divide the source of errors into two panels, the left panel

groups the lowest values of RSD and the right panel for the highest values of RSD.

Overall, $R_{eff}$ uncertainties are larger than the ones on COT for any type of error. In the left panel of **Figure** 12**,** the three fixed model parameters errors related to an incorrect estimation of the fixed parameters of the model are weak compared to the others and remain below 0.3% for mono-angular retrievals. As explained in section 4, the fixed altitude does not contribute to the uncertainty on the two retrieved parameters. The average uncertainties originating from the fixed value of $v_{eff}$ are about 0.05% for COT and slightly higher (0.15%) for $R_{eff}$ since $v_{eff}$ affects the cloud bows that are also sensitive to $R_{eff}$. Concerning the surface wind speed, the uncertainties are around an average of 0.05%.

In the right panel, for mono-angular retrieval, the measurement errors contribute to an uncertainty of about 8% on the retrieved COT and of about 13% on the retrieved $R_{eff}$. The uncertainties are reduced by factor two compared to multi-angular retrieval. The multi-angular approach leads indeed to more information available for each cloudy pixel and each additional information reduces the uncertainty on the retrieved parameters in the presence of the same 5% random noise in the measurements.

The following two groups of bars correspond to the errors introduced by the cloud homogeneous assumption used in the forward model. They are the main source of errors. For mono-angular retrieval, the assumption of a vertical homogeneous profile contributes to an uncertainty of about 16% on COT and 54% on $R_{eff}$. These uncertainties are reduced by factor four in the case of multi-angular retrieval. As discussed previously, the principal effects of 1D assumptions errors at the spatial resolution of OSIRIS come from the non-independence of the cloud columns that lead to smoothing the 3D radiative fields and increasing the heterogeneity along the line of sight (Fauchez et al. 2018). They lead to an uncertainty of 28% on COT and 45% on $R_{eff}$ when a mono-angular instrument is used.

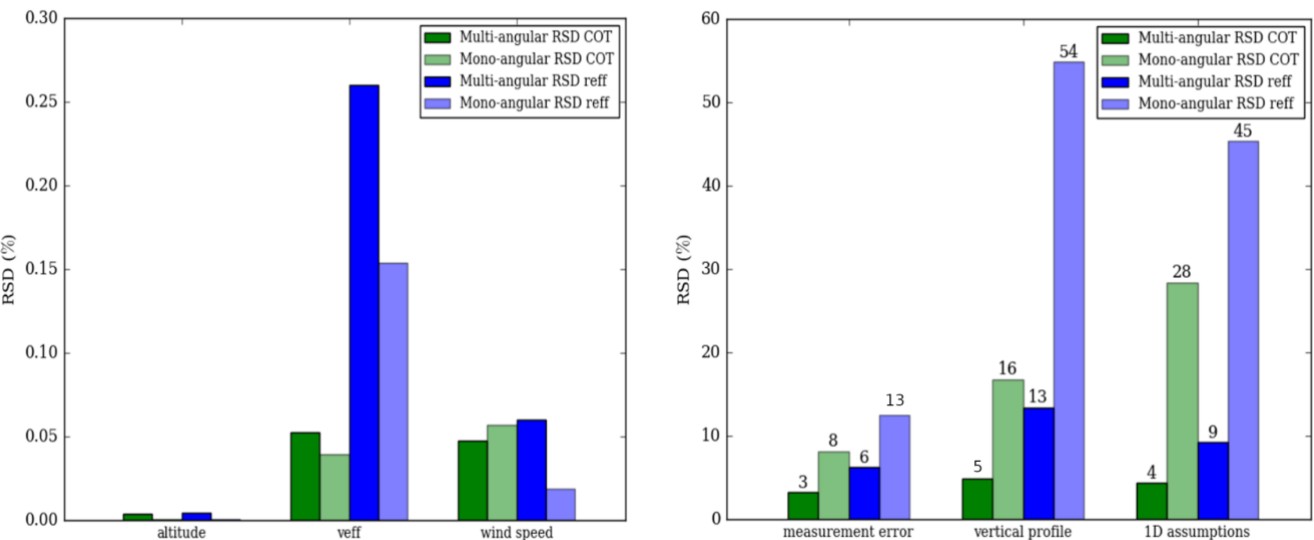

**Figure 12: Bar chart of the mean uncertainties of the retrieved COT and $R_{eff}$: green bars correspond to RSD COT and blue bars to RSD $R_{eff}$. Dark colors correspond to multi-angular retrieval and light colors to mono-angular retrieval. The errors originating from the fixed parameter errors are in the left panel and the measurements and forward model errors are in the right panel.**

The multi-angular approach provides additional information for each pixel and constrains the forward model to match all the angular radiances at once. As seen, the OSIRIS multi-angular characteristics have the advantage of decreasing

the angular effects around the cloud bow directions by adding the contribution of other geometries and mitigating the sensitivity of the retrieval issued from the assumptions in the forward model. It avoids most of the failed convergences that occurred with the mono-angular bispectral method and retrieved more homogeneous and coherent COT and $R_{eff}$
fields.

## 6 Conclusions

In this study, we present a method to retrieve two important microphysical and optical parameters of liquid clouds, COT and $R_{eff}$ and their uncertainties using NIR/SWIR multi-angular airborne measurements. The algorithm is based on the mathematical framework of the optimal estimation method (Rodgers, 2000) and focuses on assessing the different
uncertainties of the retrieved properties originating from different sources of errors.

The studied case uses the measurements of the airborne radiometer OSIRIS obtained during the CALIOSIRIS campaign. It consists of a monolayer water cloud located at 5 km altitude over the ocean with tilted solar incidence ($\theta_s$=59°).

In the first step of the retrieval, COT and $R_{eff}$ are retrieved by considering only the measurement errors (without
introducing any error linked to the forward model). The uncertainties originating from different sources of error are computed afterward by using the previously retrieved COT and $R_{eff}$ and are decomposed into 3 different sources related to (a) the instrument measurement errors, (b) an incorrect estimation of the fixed model parameters such as the ocean surface wind, the cloud altitude and the effective variance of water droplets distribution, and (c) the errors related to the vertically and horizontally homogeneous cloud assumptions. The computations are done using the multi-angular
method and for comparison a mono-angular method, which is the usual approach in the operational algorithm.

In the multi-angular retrieval, a 5% measurement error contributes to around 3% of uncertainty on the retrieved COT and 6% on the retrieved $R_{eff}$. It tends to increase with increasing values of COT and $R_{eff}$ to which the sensitivity of radiances starts to decrease. Since they are not characterized, the correlations between the measurement errors issued from different viewing angles are not considered in our retrieval, but they could increase these values. Nevertheless,
when considering a mono-angular retrieval, these uncertainties are doubled.

The uncertainties related to the fixed parameters remain low with both mono and multi-angular retrieval. The largest one is due to the unknown value of the effective variance of the droplet size and is respectively equal to 0.15% and 0.25% for the mono and multi-angular cases. Note that, since the information provided by Lidar or polarized measurements was used, the uncertainty for the non-retrieved parameters was chosen to be low. For applications to
cases without this available information, errors would be higher. If the method is applied to 3MI for example, the errors related to the cloud top altitude would be higher as the O2-A band leads to cloud top pressure uncertainties between 40 and 80hPa depending on the cloud types (Desmons et al. 2013). A more complex algorithm could also be used with a measurement vector including O2-Aband radiances and multi-angular polarized radiances to have information on and to add the cloud top altitude and the effective variance (Huazhe et al. 2019) in the state vector.
This study clearly shows that the largest uncertainty is due to the homogeneous cloud assumption made in our forward model. First, the uncertainties related to the homogeneous vertical profile were quantified using a heterogeneous LWC

profile with a triangle shape (known as quasi-adiabatic) composed of two adiabatic profiles. This more realistic profile takes into account the transition zone at the top of the cloud-related to turbulent and evaporation processes. The scene averaged values reach 5% and 13% for COT and $R_{eff}$ respectively in the multi-angular retrieval of our case study and go up to 16% and 54% for COT and $R_{eff}$ respectively when using mono-angular measurements. The largest uncertainties are obtained for the largest cloud optical thickness as the radiations sample only the higher layers of the cloud where the information is different between the homogeneous and heterogeneous vertical profiles.

The other sources of uncertainty related to the simplified cloud physical model come from the radiatively non-independence of the cloudy columns that dominate at the high spatial resolution of OSIRIS. In the optically thin overcast cloud case studied here, the scene average uncertainties originating from the 3D effects are 4% for COT and 9% for $R_{eff}$ when using multi-angular measurements, and 28% for COT and 45% for $R_{eff}$ when using mono-angular measurements. The non-independence of the cloud columns dominates and tends on one hand to smooth the 3D radiative field compared to radiances computed with the independent pixel approximation and on the other hand to increase the cloud property heterogeneity along the line of sight.

The method was applied to real data, which means that the true cloud parameters are unknown. Consequently, it is not possible to know if real errors on the retrieved parameters are included in the uncertainties given by the method presented here. One reason that can lead to an erroneous assessment is that the estimations of the uncertainties are done around retrieved values than can be biased. A way to check the consistency of the method and the validity of the uncertainty ranges would be to simulate radiances using Large Eddy Simulation model with realistic cloud physical description, add noise for the errors measurements and derive the cloud parameters and their uncertainties.

The method presented here can be adapted to the future 3MI imager. The first step which consists of including the uncertainties related to the measurement errors is directly implementable in an operational algorithm. The second step which consists of computing the uncertainties resulting from the non-retrieved parameters is more computationally expensive but could also be included. The uncertainties related to the non-retrieved parameters, in addition to the one related to measurement errors, have already been implemented since Collection 5 in MODIS operational algorithm through the computation of covariance matrix where Jacobians are derived from look-up table and completed for Collection 6 (Platnick et al. 2017). Concerning the forward model errors, the method cannot be implemented as in this work, in an operational algorithm because of the prohibitive computation time but a climatology based on several cases studies, depending on the type of clouds, land or ocean surface flag for example could be used in order to obtain a distribution of the errors according to the scene characteristics.

The results obtained in this study show, not surprisingly regarding the numerous studies already published, that the vertical and horizontal homogeneity assumptions are major contributors to the retrieval uncertainties. One way to reduce it would be to define a more complex cloud model that can take into account the vertical and horizontal heterogeneity. This adds more complexity to the forward model as it would imply to retrieve more sophisticated cloud parameters (e.g. extinction or effective size profile). It appears however possible given the important and complementary information provided by OSIRIS or 3MI measurements. Recent studies proposed to retrieve vertical

profile using cloud side information (Ewald et al., 2018; Saito et al., 2019, Alexandrov et al., 2020) or to realize multi-
pixel retrieval to account for the non-independence of the cloudy pixels (Martin et al., 2014; Okamura et al., 2017; Levis et al., 2015) and their implementation could be studied.

**Code and data availability**

Data and codes are available upon request to the authors.

**Author contributions**

CM developed the inversion algorithms and performed the analysis of the results with the support of CC, FP and LCL. FA and FP participated in the CALIOSIRIS airborne campaign. FA and JMN made the calibration and OSIRIS data processing from level 0 to level 1. All authors discussed the results and contributed to the final manuscript.

**Competing interests**

The authors declare that they have no conflicts of interest.

**Acknowledgments**

The authors thank Guillaume Merlin and Anthony Davis for fruitful discussions concerning the cloud vertical heterogeneous profile. This work has been supported by the Programme National de Télédétection Spatiale (PNTS, http://www.insu.cnrs.fr/pnts), grant n° PNTS-2017-03 and by the CPER research project CLIMIBIO. The authors thank the French Ministère de l'Enseignement Supérieur et de la Recherche, the Hauts-de-France Region and the European
Funds for Regional Economical Development for their financial support to this project.

Airborne data were obtained using the aircraft managed by Safire, the French facility for airborne research, an infrastructure of the French National Center for Scientific Research (CNRS), Météo-France and the French National Center for Space Studies (CNES).

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
