# Peer review of "Liquid Cloud Optical Properties Retrieval and Associated Uncertainties using Multi-angular and Bispectral Measurements of the Airborne Radiometer OSIRIS"

_Atmospheric Measurement Techniques, 2021_

## Referee Comment (RC1)

Manuscript number: amt-2021-414
Full title: Cloud optical properties retrieval and associated uncertainties using multi-angular and multi-spectral measurements of the airborne radiometer OSIRIS
Author(s): Matar et al.

The paper investigates the retrieval uncertainties of liquid cloud properties, specifically cloud optical thickness (COT) and droplet effective radius (Reff), associated with various error sources using an optimal estimation (OE)-based retrieval procedure. The retrieval error covariance matrix is decomposed into the one originating from individual sources of errors, which makes it possible to evaluate the quantitative retrieval uncertainty estimation for each error source, including measurement error, model parameter errors, and error due to the assumptions in the forward model. The authors apply this framework to multi-angular and multi-spectral measurements made by OSIRIS airborne radiometer with a focus on liquid clouds. The results indicate the forward model assumptions that do not consider in-cloud heterogeneous profiles and 3D radiative transfer effects induce the largest uncertainties in the retrievals, followed by the measurement errors. While the model parameter errors provide the least impact on the retrieval uncertainties that are less than 0.5% of the retrieval quantities. These error estimates and retrieval procedures will be useful for the further 3MI observations.
While this paper is well written in the introduction and methodology part, I am concerned about the representativeness of the results and the analysis flow of this work. Please find my comments below. The topic presented in this paper is suitable for Atmospheric Measurement Techniques. I recommend Major Revisions to reconsider the manuscript for publication.

**Major comments**

*Representativeness*
        First of all, the paper discusses only the relative standard deviations (RSD) of the retrieval variables. Why don't the authors discuss the retrieval _biases_ associated with the measurement errors, model parameters, and assumptions in the forward models? In particular, the vertical heterogeneity and 3D radiative effects would induce substantial biases in the retrievals. The different magnitude of errors among observational bands and angular directions would cause impacts on the cloud property retrievals (through changing the sensitivity weights). However, the relative standard deviations of the retrieval variables do not tell you how much the retrieval variables are biased. If the authors want to solely focus on the retrieval RSD for each error source, then it can be more appropriately achieved through numerical experiments based on synthetic simulations of observational signals based on this framework or through incorporating the root-mean-square error (RMSE) that includes the retrieval bias information and can be theoretically derived from the bias and RSD. Alternatively, the authors may consider to additionally perform cloud property retrievals based on error covariance matrix that considers ALL sources of errors (i.e., measurement errors, model parameter error, and forward model errors) to see how different the cloud property retrievals are compared to measurement error-only cases.
        In addition, the uncertainties and fixed parameters of the source of errors seem to be determined in different ways. The cloud top height value and its uncertainty are derived based on collocated lidar observations (i.e., representing the CALIOSIRIS campaign), the effective

variance seems to be arbitrarily determined, and the wind speed value and uncertainty may represent global statistics. This gives me a question on the representativeness of this study. If the authors make the results useful to the future 3MI mission, the uncertainty (specifically, the standard deviation of a model parameter) of each error source (in particular, the model parameter errors) should represent those of global climatology. Specifically, I feel that the cloud top height and effective variance uncertainties may be a bit too small, so the retrieval RSD might be underestimated (although the general conclusion the little contribution of model parameter uncertainties to retrieval RSD may not change).

For these reasons, I suggest the authors clearly state what are the focuses of this paper in the last paragraph of the introduction and reconsider the experimental design of this work through, but not limited to, using numerical experiments, adding the retrieval biases to the current analysis, or other appropriate methods.

*Analysis flow*

While I appreciate the authors for describing the solid mathematical basis of the analysis procedure, I may have an argument on the use of COT and Reff retrievals obtained based only on measurement noise in the error covariance matrix for the following analysis. This can overfit the retrieval variables to obtain an optimal solution (i.e., $J \sim n_y$) because radiative signal *perturbations* (here what I mean is any of radiative perturbation induced by atmospheric-cloud properties naturally occur in the real world) originating from other sources of errors are partly explained with the *biased* retrieval values. If such perturbations are significant (and yes, it is significant particularly for vertical heterogeneity and 3D effects), then the sensitivity of the other error sources to the retrieval quantities are obtained from the sensitivities to the retrieval variables at the biased cloud property conditions. Ideally, numerically generated cloud property fields (such as Large Eddy Simulations) would provide datasets for the evaluations of the retrieval uncertainties based on the authors' flamework. As the true cloud properties are not available in the observed cloud field, it is not possible to address this issue based on the given observations. Therefore, at least, the authors should discuss a potential bias of the retrieval uncertainty evaluations based on this framework.

**Minor comments**
1. Title: Actually, the authors use only two bands for the cloud property retrievals, and therefore, "bispectral" measurements would be more descriptive rather than "multi-spectral" measurements?

2. Line 241, Eq. (7): Why do the authors use a linear scale of the cloud optical thickness for a state vector element, not a logarithmic scale? Although the relation between solar reflectance and COT is linear at very small COT conditions ($\tau \ll 1$), it is in general quite non-linear over most of the COT range, which makes the retrieval process slow and may degrade the convergence, and also may limit the representativeness of this results to optically thin cases.

3. Section 3.3: I do not find the values of *a priori* error covariance matrix in the manuscript. Please briefly mention what values are chosen.

4. Line 338 "$\sigma_{alt} = 0.16$" :: This represents only the cloud properties observed during the CALIOSIRIS campaign.

5. Line 333 "$\sigma_{veff} = 0.003$" :: How did you get this value? It seems too small. Please site references that support this quantity.

6. Lines 372–376: In addition, the collision-coalescence process can provide a larger droplet radius at the lower part of clouds, which are observed from CloudSat. As vertical heterogeneity is important to the cloud property retrieval, the authors may consider a better representation of cloud profiles using a better cloud profile parameterization (e.g., Saito et al., 2019).

7. Lines 416–420: As the authors assume the flat cloud top, which reduces some of cloud 3D effects such as illuminating and shadowing effects, I had an impression that the authors may focus on lateral photon transport effect here. If so, it would be better to rephrase 3D with lateral photon transport or state "3D" regarded as the lateral photon transport.

8. Figure 4: The readers cannot recognize if there are optimal/non-optimal solutions from these plots. If it is non-optimal, a set of cloud retrievals may not adequately explain the measured signals. I suggest the authors add the cost function distributions in addition to these two plots.

9. Lines 483–484 *"3D effects due to solar illumination do not appear in the retrieved cloud properties"* :: This is an obvious statement as the cloud top is assumed to be flat, which removes the shadowing and illuminating effects. Please state that this error evaluation focuses solely on the lateral photon transport effect.

10. Lines 539–540: The cost function divided by the number of measurement signals ($J/n_y$) is a comparable quantity among mono-angular- and multi-angular-based retrievals.

11. Lines 559–560" Why are these uncertainties reduced for multi-angular cases?

12. Line 610: If the authors state that *"the uncertainties related to the measurement errors is implementable in an operational algorithm,"* the uncertainty evaluations should be based on global climatology of cloud, surface, and atmospheric properties. A limited case (i.e., a granule of a cloud scene) may not be adequate to state so.

13. Typo and grammatical errors: Please proofread the main body of the manuscript again. I have found several grammatical errors, e.g., Lines 557, 571, and possibly more.

**Reference**

Saito, M., P. Yang, Y. Hu, X. Liu, N. Loeb, W. L. Smith Jr., and P. Minnis, (2019) An efficient method for microphysical property retrievals in vertically inhomogeneous marine water clouds using MODIS–CloudSat measurements, *J. Geophys. Res. Atmos.*, 124, 2174-2193.

---

## Referee Comment (RC2)

Review of 'Cloud optical properties retrieval and associated uncertainties using multi-angular and multi-spectral measurements of the airborne radiometer OSIRIS'

By Christian Matar et al.

**General comments**

This paper describes an approach for the retrieval of cloud optical thickness and droplet effective radius from multi-angular and multi-spectral satellite measurements. In addition, estimates of the retrieval errors due to different error sources are presented. The main novelty is an attempt to estimate retrieval errors caused by deviations from vertically homogeneous profile and independent pixel assumptions. The topic is important and suitable for this journal. Overall the retrieval setup appears to be sound and the results are plausible. However, I do have some general concerns and questions that need to be addressed before this paper can be published.

- Only one case is studied, which limits the validity of the results. Retrieval errors, in particular those caused by deviations from homogeneous cloud and independent pixel assumptions, will depend on the scene. While discussing more cases may be outside the scope of the paper, these limitations at least need to be mentioned.
- The paper gives a false impression of the current cloud optical and microphysical property algorithms. Firstly, most of these do consider measurement errors and produce retrieval error estimates (e.g., Platnick et al., 2017), unlike what is written in the abstract. Secondly, for many of these algorithms the total retrieval error has been separated into contributions from individual error sources (e.g., Walther and Heidinger, 2012), unlike what is written on page 10.
- It is not clear why polarization measurements have not been included in the retrieval. It looks like these could, amongst others, further constrain the width of the size distribution, which has instead been fixed prior to the retrievals.
- There are many textual mistakes and inaccuracies, of which only some examples have been included in this review.

**Specific comments**

Title: The paper only deals with liquid clouds. This must be included in the title.

Abstract, L13: Clouds are really characterized by more properties than these two, e.g. the height of the top and base.

Abstract, L16: As mentioned above, measurement errors are considered in most retrieval algorithms.

Abstract, L26: I suggest not to refer to 'traditional' bi-spectral retrievals as 'MODIS-like methods' since these methods are not in any way specific to MODIS.

P1, L36: The statement is still true so why not cite the latest IPCC report here?

P2, L40: Measurements of emitted radiation are also important for cloud property retrievals.

P2, L50: Suggest to cite the more recent Platnick et al. (2017).

P2, L51: The bi-spectral method is applied to many instruments, not 'in particular' to MODIS.

P3, L93: What is 'roughening of the radiative field'?

P4, L110/111: LUT-based retrieval algorithms are very well suited to produce retrieval error estimates. Please remove this statement.

P4, L125: 'model': do you mean 'state'?

P5, Fig. 1: The figure and caption can be improved. Lines cannot be distinguished very well. What are 'optical matrices'? Spectral response functions is probably a more common term. And in what units are these plotted / how have they been scaled? The y-axis only says 'Transmittance'.

P5, L163-165: Can you expand a bit on the viewing angles? Here numbers of 19 and 20 angles are mentioned, later it is 13 angles. How many angles were actually used for the retrievals? Can you also comment on how well the pixels for the different viewing angles are aligned? And if they are aligned at ground level, isn't there a spatial mismatch at cloud altitude?

P6, L175: What is 'LIDAR-LNG'? And what is 'the vertical profile' in Fig. 2b? There is a whole series of vertical profiles.

P6, L179: There are some 'intense white' but also quite dark parts in the reflectance image. Please explain what you mean.

P6, L182: Where do you see a 'white arc' in Fig. 2d? I only see an arc with rainbow colors.

P6, L185: Could you perhaps indicate in the figure where the reflectance is (or may be) affected by sun glint?

P7, Fig. 2 and caption: Please use UTC (not local time) throughout to avoid confusion. What is the background in Fig. 2a? Please add explanation of red arrow in Fig. 2a and red rectangle in Fig. 2b in the caption and not (only) in the text. What is the half-visible text on top of that image? What are all the colors in the legend (most of which do not appear in the figure)? What is the blue bar on the left-hand side of Fig. 2c? I don't think that 490-670-865 nm is a 'true-color' RGB composite. From which viewing angle are these radiances? The contours are not concentric.

P8, L205: It is stated that vector y has dimension $n_y$ but in Eq. (8) it has dimension $2*n_y$.

P8, L218: Please state explicitly how $x_a$ and $S_a$ are defined ('large' is too vague).

P8, L225: Suggest to refer to page 10 where the Jacobian is explained.

P9, L250-251: Is this justified? From Fig. 1 it looks like there is considerable absorption, in particular in the 2.2 micron channel, which needs to be accounted for.

P9, L255: What are these 'measurements'?

P9, L258: What is 'independent column approximation'? Is it the same as the 'independent pixel approximation' introduced on page 2? If so, please use consistent terminology.

P9, L264-268: How do you calculate the Jacobian matrix?

P10, L274: The off-diagonal terms are non-zero so it is confusing that they have been written here as zeros. Can you explain?

P10, L276: Do you mean Eq. (7) instead of Eq. (10)?

P10, L287: As mentioned in the general comments, there have been other studies where the types of error were separated.

P11, L300: There are other ways of calibration, e.g. vicarious.

P11, L301: I would argue that calibration usually addresses systematic (not random) errors in the measurements.

P11, L304: According to Eq. (8) the dimension of the measurement vector is 2 n_y x 2 n_y (?)

P11, L306: Do you have a motivation for taking a fixed 5% measurement error?

P12, L341: This value of v_eff is quite small. Could you add some explanation on how it is determined? Wouldn't it be possible to include its determination in the overall retrieval (by including the polarization measurements in the observation vector)?

P12, L336-343: In this case the cloud top height (and thickness) and effective variance can be determined very accurately. However, in 'real life' uncertainties will me much larger (e.g., if you have no lidar available). Wouldn't it therefore be better to work with larger uncertainties so that the resulting error estimates become more representative?

P13, L355-358: In the OE framework errors are assumed to be Gaussian and error estimates reflect 1-sigma of the Gaussian distributions. Can you comment on the Gaussian nature of the forward model related errors? Is it plausible to use the difference between two configurations as 1-sigma of the uncertainty, or would these configurations rather reflect two extremes?

P13, L358: What is the square of a matrix?

P13, L363: To estimate retrieval errors due to deviations from the assumption of vertical homogeneity, a specific alternative cloud model is outlined in detail. However, it should be realized that this is just one possibility. For example, real profiles have a varying degree of sub-adiabaticity, which is not considered here. What would be the effect on the uncertainty estimates?

P14, Fig. 3: The cloud is placed between 5 and 6 km. Where do these numbers come from? They are not the same as on page 12. Please include the settings (top and bottom height, cloud optical thickness, maximum effective radius, …) for this particular figure in the caption.

P15, L396-398: Is it correct to determine the maximum effective radius such that the average effective radius of the heterogeneous and homogeneous profiles are the same? Shouldn't R_eff be weighted with extinction? Or, alternatively, a requirement to arrive at the same liquid water path for both profiles seems better justified.

P15, L407: Only the IPA seems to be addressed here. What about the PPH assumption?

P15, L409-411: It seems to be stated that the PPH assumption includes the IPA, whereas in earlier parts of the manuscript they were introduced as different things (which I think they are).

P16, Fig. 4: Does the scene contain clear-sky pixels? If so, how are they reflected in the COT and Reff maps? Are there any failed retrievals? If so, how are they reflected in the maps?

P16, caption Fig. 4: Is the date a typo or is this really a different case from the one introduced in Fig. 2?

P16, L436: A figure with COT and Reff uncertainties as functions of COT and Reff would be very instructive to illustrate this.

P16, L435: The uncertainties in COT appear to be very low. Is a retrieval error of 0.5% realistic? For thin clouds COT depends approximately linearly on the reflectance. How can a reflectance

measurement uncertainty of 5% result in an order of magnitude lower uncertainty in COT? Is this thanks to the combined information from different viewing angles. But, if that's the case, isn't the assumption of uncorrelated errors between the measurements from these different angles much too optimistic?

P17, 459-460: Could this estimate be too optimistic? In case of broken clouds, sun glint can have a relatively much higher impact on the measured reflectance, which would not be captured here.

P18, L474: For COT it seems to be rather something like 8 %.

P19, L481-484: This is a firm statement, for which no evidence is provided.

P19, L485-486: There are no sub-pixel measurements, and sub-pixel cloud variability is not represented in this work. Again, this is a statement without proof. A PPH error estimate should be added to the retrieval setup, so the PPH effect can be quantified.

P20, Fig. 8: How is radiance defined here? Is it the sun-normalized radiance or true reflectance? From which of the 13 viewing angles are these measurements taken?

Fig. 9: Nice figure, illustrating the different response of thin and thicker cloud portions to 1D versus 3D radiative transfer.

P21, L508-509: Is the nearest-nadir view used for the mono-angular retrievals?

P21, Fig. 10: I am shocked by the enormous differences between the mono- and multi-angular retrievals. Ok, for the cloud bow geometries it is well known that mono-angular retrievals do not work. However, for other geometries the mono-angular retrieval should give a reasonable solution, in particular for a reasonably 'well-behaved' cloud field as studied here. This asks for further clarification. Can you also include a scatter-density plot comparing COT and Reff from the two retrievals on a pixel basis?

P22, L530-531: Apparently both retrievals fail to converge in some cases. But there do not seem to be missing values in Figs. 4 and 10. How can that be explained? What output does the algorithm give in case of no convergence? Are these cases included in the statistics? Are statistics based on a common set of mono- and multi-angular successful retrievals?

P22, L531-532: Is the multi-angle retrieval expected to retrieve smaller Reff? Can you explain that? And why would smaller Reff lead to lower COT?

P22, L533-534: This is not true. The measurement pair can be outside the 2D LUT space (and I guess this is what happens in the reported 5.9% cases of failed convergence).

P24, Fig. 12: The decrease in retrieval error from mono- to multi-angular retrievals is spectacular, especially with respect to the vertical homogeneity and IPA assumptions. Can you explain in some more detail how that is achieved? Still, differences between the two retrievals (Fig. 10 vs. Fig 4) appear (much) larger than accommodated by the respective error estimates. Can you comment on that?

P24, Fig. 12: The mean Reff retrieval error due to measurement errors is 12.55 in Fig. 11 but 12 in this figure, which is not consistent.

P25, L596: In Fig. 12 the mean COT error is 4%, not 5%.

P25, L605-607: Please remove since this was not shown (or alternatively include in the retrieval error estimates).

**Technical corrections**

P1, L22: Acronyms (POLDER in this case) must be written out.

P1, L16/L17: '… without considering … the choice of ancillary data': What does it mean that the choice of ancillary data is not considered?

P1, L31: 'uncertainties on': should be 'of'. Occurs frequently throughout. Please correct.

P2, L53: The second sentence does not follow from the first, so the word 'Therefore' is misplaced.

P3, L76: increase -> increasing

P3, L80: 'radiations' is not really a word.

P3, L90: by its -> in

P3, L96: vertical -> vertically

P4, 113: Usually, the acronym is put between brackets after the full name instead of the other way round.

P4, L124: Bayesian (with capital)

P9, L242: Add $\lambda_a$ and $\lambda_b$ after wavelengths.

P9, 239: Italic case is not needed here (similar occurrences throughout).

P9, 240: Variables in italic ($R$ in $R_{eff}$ should be italic). (similar occurrences throughout).

P9, L243: (8) is duplicated.

P9, L250: 'All the' -> 'the two'?

P10, L271: 'implantation': do you mean 'implementation', 'inclusion', ..?

P10, L271: adjust -> adjusts

P10, L306: 'measurement errors that cover the measurement errors'?

P11, L13: Italics appearing here and there are not needed and confusing.

P12, L327: Should (17) and (18) be reversed?

P12, L328: Should this be $K_{b\_i}$ instead of $K_i$?

P12, L340: for -> to

P14, Fig. 3: Minus sign in the x-axis label is confusing.

P15, L395: exctinction -> extinction

P15, L417: minimized -> underestimated (?)

First paragraph on page 16: here I give a more complete inventory of textual mistakes as guidance for the rest of the manuscript.

P16, L422: Both bispectral and bi-spectral occur in the manuscript.

P16, L423: weak -> weakly

P16, L423: .. channel partially absorbed by ..: how can a channel be absorbed?

P16, L424: on -> to

P16, L424: Remove 'up to' (?) I guess all viewing angles are available. By the way, does this mean that n_y = 13?

P16, L425-426: 'This error is straightforward': how can an error be straightforward?

P16, L429: ertically -> vertically

P17, L443: As noted before, do not write variables like COT, and mathematical operations like RSD, in italics.

P17, L457: 'enlarge the directions': what does that mean?

P19, caption Fig. 7: 'model' missing after 'forward'?

P19, L478: What are 'these differences'?

P21, L503: assumption -> assumption

P21, Fig. 10: For comparability with Fig. 4 it would be good to use the same color scales. Can you also add the mean values? Also, add some whitespace between the maps and the color bars.

P22, caption Fig. 11: Add 'angle' after 'scattering'.

P22, Fig. 11: Is this figure for the mono-angular retrieval?

P23, L542: spatially -> spatial

P23, L557: 'to the' is duplicated.

P23, L557: what is a 'homogeneous assumption'?

P24, L571: 'retrieve' is duplicated.

P24, L583: horizontal -> horizontally, vertical -> vertically

P25, L587: for -> to

P25, L590: what is 'miss-knowledge'?

References: Journal names are missing in all references.

**References**

Platnick, S., Meyer, K. G., D., K. M., Wind, G., Amarasinghe, N., Marchant, B., Arnold, G. T., Zhang, Z., Hubanks, P. A., Holz, R. E., Yang, P., Ridgway, W. L., and Riedi, J., 2017: The MODIS Cloud Optical and Microphysical Products: Collection 6 Updates and Examples From Terra and Aqua, IEEE T. Geosci. Remote, 55, 502–525, doi: 10.1109/TGRS.2016.2610522.

Walther, A. and Heidinger, A. K., Implementation of the daytime cloud optical and microphysical properties algorithm (DCOMP) in PATMOS-x, J. Appl. Meteorol. Climatol., 51, 1371-1390, doi:10.1175/JAMC-D-11-0108.1.

---

## Author Comment (AC3)

[revised manuscript text omitted]

Therefore, it can be easily shown that the best estimate of the state vector $x$ corresponds to the minimum of the so-called cost function $J(x)$:

$$J(x) = [y - F(x)]^T S_\epsilon^{-1} [y - F(x)] + [x - x_a]^T S_a^{-1} [x - x_a] \tag{3}$$

The first term of $J(x)$ represents the difference between the measurements and the forward model calculated for a given state vector $x$, weighted by $S_\epsilon$ the covariance matrix associated with the measurement error and the forward model. The second term represents the difference between the state vector $x$ and the a priori state vector $x_a$ weighted by $S_a$ the covariance matrix associated with $x_a$. In line with the cost function, the optimal  estimation emerges from a balance between the information carried by the measurement about the state vector and what we already know about it before the measurement. In our case, we do not have a prior estimate of the state vector. The iterations are initiated by a first guess while applying a large $S_a$. The difference between the measurements and the forward model

280     will be the decisive element in the minimization of the cost function. It will ensure that the estimated cloud properties have the optimal fit with the observed system only.

    The minimization is done through the Levenberg-Marquardt approach (Marquardt, 1963; Levenberg, 1944) based on the "Gauss-Newton" iterative method. Assuming the model is nearly linear around a given state vector, each iteration is calculated following the Eq. (4):

285
$$x_{i+1} = x_i + S_{x_i}^{-1}\left[ K_i^T S_\epsilon^{-1}\left( y - F\left(x_i\right)\right) - S_a^{-1}\left( x_i - x_a\right)\right] \tag{4}$$

    where $x_i$ is the state vector at the $i_{th}$ iteration, $K_i$ is the sensitivity (or Jacobian) matrix described in Eq. (10) and $S_{x_i}$ is the covariance matrix of the state vector defined in Eq. (5).

$$S_{xi} = \left[\left(1+\gamma\right)S_a^{-1} + K_i^T S_\epsilon^{-1} K_i\right]^{-1} \qquad \underline{\quad\quad} \tag{5}$$

    The parameter $\gamma$ affects the size of the step at each iteration. If the cost function increases at an iterative step $i$ then $\gamma$ is increased and a new smaller step ($x_{i+1}$) is calculated until the cost function decreases.

290     The iterative process stops when the simulation fits the measurement (Eq. (6)), named convergence of Type 1 or when the iteration converges (Eq. (7)) named convergence of Type 2. The left side of Eq. (6) represents the normalized cost function without taking into account the a priori negligible contribution. When the cost function  is smaller than $n_y$ or the normalized cost function ($J/n_y$) less or equal to one,  the iterations stop. Eq. (7) deals with the iterative steps and will make sure that the iterations will stop when the difference between two

295     successive steps weighed by $S_x$ is less than $n_x$. In other words, when further changes in the state vector have small to zero changes in the minimization.

$$\left[y - F\left(x_i\right)\right]^T S_\epsilon^{-1}\left[y - F\left(x_i\right)\right]/n_y \le 1 \
[revised manuscript text omitted]

---

## Author Response (AR1)

**Reply To RC1:**

We are deeply grateful to the reviewer for their constructive and pertinent comments on the manuscript, which help to greatly improve the paper.
Please find hereafter our point-by-point responses to the comments and suggested corrections.
Comments are in black, our responses in blue and the text modifications in green (in bold when only one part of the sentence was modified).
Note that the indicated line numbers correspond to the line numbers of the "track changes" version.

RC1:
Manuscript number: amt-2021-414
Full title: Cloud optical properties retrieval and associated uncertainties using multi-angular and multi-spectral measurements of the airborne radiometer OSIRIS
Author(s): Matar et al.

The paper investigates the retrieval uncertainties of liquid cloud properties, specifically cloud optical thickness (COT) and droplet effective radius (Reff), associated with various error sources using an optimal estimation (OE)-based retrieval procedure. The retrieval error covariance matrix is decomposed into the one originating from individual sources of errors, which makes it possible to evaluate the quantitative retrieval uncertainty estimation for each error source, including measurement error, model parameter errors, and error due to the assumptions in the forward model. The authors apply this framework to multi-angular and multi-spectral measurements made by OSIRIS airborne radiometer with a focus on liquid clouds. The results indicate the forward model assumptions that do not consider in-cloud heterogeneous profiles and 3D radiative transfer effects induce the largest uncertainties in the retrievals, followed by the measurement errors. While the model parameter errors provide the least impact on the retrieval uncertainties that are less than 0.5% of the retrieval quantities. These error estimates and retrieval procedures will be useful for the further 3MI observations.
While this paper is well written in the introduction and methodology part, I am concerned about the representativeness of the results and the analysis flow of this work. Please find my comments below. The topic presented in this paper is suitable for Atmospheric Measurement Techniques. I recommend Major Revisions to reconsider the manuscript for publication.

**Major comments**

Representativeness
First of all, the paper discusses only the relative standard deviations (RSD) of the retrieval variables. Why don't the authors discuss the retrieval biases associated with the measurement errors, model parameters, and assumptions in the forward models? In particular, the vertical heterogeneity and 3D radiative effects would induce substantial biases in the retrievals. The different magnitude of errors among observational bands and angular directions would cause impacts on the cloud property retrievals (through changing the sensitivity weights).
However, the relative standard deviations of the retrieval variables do not tell you how much the retrieval variables are biased.

The work presented in this paper is based on the optimal estimation method. The principle of the method is to determine the most probable state knowing that measurements have been performed with uncertainties represented by the PDF of the measurements. The reasoning is based on the Bayesian formalism with a-priori and a-posteriori probability densities which allow linking the space of the states to the space of the measurements. The retrieved state is then the most likely consistent state according to the available information. In the optimal estimation method, all the PDF are assumed to be Gaussian PDFs. This implies that the uncertainties can be estimated by the covariance matrix of the parameters, and then by the relative standard deviation (RSD).

However, we agree with the reviewer that the use of Gaussian PDFs can have an impact in the assessment of the uncertainties depending on the type of errors : the measurements errors are usually, at first order, considered as random errors. They can consequently be modeled by a Gaussian distribution. The non-retrieved parameters and the forward model parameter errors are derived following Rodgers, 2000 (chapter 3). The computation of these two errors lies on the assumption that the forward model can be locally linearized about the state vector value. If the non-retrieved parameters have been properly retrieved, their own uncertainties are unbiased and can be modeled by a Gaussian PDF. With the assumption of a linear model, the resulting errors on the state vector are unbiased and can also be represented by a Gaussian PDF.

The forward model error is obtained (Eq. 21) from the difference between the results of the simplified model F used for the retrieval and the realistic model F' multiplied by the Gain matrix, which represents the sensitivity of the retrieval to the measurements. A bias between the results of the two models will then be included in the Gaussian pdf representing the forward model errors, which tends to overestimate the errors.

To clarify these different points and clearly state the different assumptions made. We add several paragraphs or sentences:

In section 3.1, we add a paragraph explaining the Bayesian formalism and the assumption of the Gaussian distribution, line 25 :

"To achieve it, a Bayesian probabilistic approach is applied. Before the measurements, an a priori knowledge of the state vector can be described by a probability density function (PDF). Once the measurements have been carried out, this knowledge can be described by the posterior PDF of the state, which is a conditional probability (probability of having given that is true). The posterior PDF of the state vector can be related to its a priori PDF by the Bayes' theorem:

$$P(x|y) = \frac{P(y|x).P(x)}{P(y)} \tag{2}$$

Where P(y) is the PDF of the measurements including the uncertainties and P(y|x) is the PDF of the measurements given that we know the state vector.

In the optimal estimation method, the previous PDFs are represented by Gaussian distributions, assuming that the errors of the measurements, the errors related to the non-retrieved parameters and the errors of the forward model are normally distributed around a mean value. In other words, we assume that the model can be linearized around the most probable state vector.

We also add clearly the assumption of linearity, section 3.1, line 280 :
Assuming the model is nearly linear around a given state vector,...

The choice of RSD to characterize the errors derives from the assumption of Gaussian distribution. We specify line 350 :

**The use of Gaussian PDF** leads to compute the uncertainty on a particular parameter $x_k$ as the square root of the corresponding diagonal elements **of the covariance matrix** $\sigma_k = \sqrt{S_{xkk}}$, where k is the index of the parameter in the state vector x.

We add later in section 3, a sentence for each errors indicating that we assume a Gaussian distribution and linearity of the model about the state vector value :

- Line 381, for the measurements errors: "The **uncertainties** of the measurements **remaining after the calibration processes are assumed**, random, uncorrelated between channels and **can be consistently approximated as a Gaussian** probability density function over the

measurement space"

- Line 400, for the errors related to the non retrieved parameters: "These errors are considered to be independent and random under the assumption of linearity of the radiances around the non retrieved parameters "
- Line 448 , concerning the errors related to the forward model: "The simplified model used for the retrieval can lead to biased retrieved parameters. In this case, the bias will be included in the Gaussian PDF width, resulting in an overestimation of the uncertainties."

If the authors want to solely focus on the retrieval RSD for each error source, then it can be more appropriately achieved through numerical experiments based on synthetic simulations of observational signals based on this framework or through incorporating the root-mean-square error (RMSE) that includes the retrieval bias information and can be theoretically derived from the bias and RSD.

The aim of our paper is not to give an exhaustive view of the possible errors concerning optical thickness and effective radius retrieval. It is to present a method to derive, from real data separately the different sources of uncertainties and to evaluate them in one example. The idea is thus not to have a general representativeness of all the possible cases, which is a huge work but more to focus on an airborne data campaign and shows how from multi-angular measurements, it is possible to derive the usual optical thickness and effective radius parameters with the partitioning of their uncertainties. To clarify our objectives, we add, as mentioned later, a paragraph at the end of the introduction section (line 141-146).

Alternatively, the authors may consider to additionally perform cloud property retrievals based on error covariance matrix that considers ALL sources of errors (i.e., measurement errors, model parameter error, and forward model errors) to see how different the cloud property retrievals are compared to measurement error only cases.

We agree that the method suggested by the reviewers is the ideal way to assess the uncertainties of the retrieved parameters but, in reality, it is impossible to implement for computational cost reasons because 3D RT simulations are much too long.
In addition as the uncertainties due to the forward model are  large, the optimal estimation algorithm will tend to converge at the first iteration as the simulated radiances will be included in pdf defined by their measurements and covariance matrix.

In addition, the uncertainties and fixed parameters of the source of errors seem to be
determined in different ways. The cloud top height value and its uncertainty are derived based on collocated lidar observations (i.e., representing the CALIOSIRIS campaign), the effective  variance seems to be arbitrarily determined, and the wind speed value and uncertainty may represent global statistics. This gives me a question on the representativeness of this study.

As explained above, the study does not aim to be representative of all cloudy situations. The fixed parameters and their uncertainties are chosen according to the experimental set-up of the airborne campaign. During the CALIOSIRIS campaign, a Lidar was on-board the airborne, which allowed us to have an accurate value with low uncertainties for the cloud top. Obviously, if less accurate information is available, the uncertainties due to cloud top retrieval could be higher.

The effective variance was chosen according to the number of supernumerary bows visible in the polarized measurements. At the time of the CALIOSIRIS campaign, polarized radiances had calibration issues and were not usable for retrieval. They were, nevertheless, used to find the effective variance that best fits the number of the supernumerary cloud bows. Figure 1 represents  the averaged polarized radiances for a transect obtained with OSIRIS and simulated radiances using an effective variance of 0.02. As the used effective variance was based on polarized radiances of the measurements, we decided to assume weak uncertainties for Veff, by choosing a standard deviation of 15% .

[Figure]

Figure 1 (not included in the paper) : Averaged polarized radiances measured by OSIRIS for a transect in the middle of the central image of CALIOSIRIS scene and simulated polarized radiances with an effective variance of water droplet distribution equal to 0.02 (in blue), as a function of the scattering angles.

Concerning the wind speed, as no direct measurements were available, we used the value given in the NCEP reanalysis of the National Oceanic and Atmospheric Administration (NOAA) database for the day of the campaign. As we do not know the uncertainties of this value, we chose to use 10% of uncertainties for the wind speed, a value that can be adjusted if the real uncertainty value becomes known.

To clarify the choice of the fixed parameters, a sentence line 418:
"The values and the uncertainties of the fixed parameters are chosen according to the experiment setup of the campaign."

For the cloud top altitude, we modified the paragraph line 420 to 425 (modifications are in bold):
"To estimate the uncertainties originating from the fixed cloud altitude, we used **the opportunity of having the LIDAR-LNG on board the aircraft, which gives** the backscattering altitudes signal obtained around the case study of CALIOSIRIS. From 11:01:06 to 11:03:06 (time where the same cloud scene is apparent), it varies between 5.57 and 5.73 km in our cloud scene. **For practical reasons due to the radiative transfer code, we use a value of 6 km for the cloud top altitude with** a standard deviation of $\sigma_{alt} = 0.16$ km (3% of the cloud altitude). **This value is low thanks to the knowledge provided by the Lidar.**"

For the choice of effective variance, we just complete the following sentence, line 428 : "...., we fixed a value of 0.02 based on the **number** of supernumerary bows in the polarized radiances **(not shown)"**

and add line 430. As the value of $V_{eff}$ was fixed using the polarization measurements of OSIRIS, this uncertainty is weak and not representative of all clouds"

We add also in the analysis of the non retrieved parameters, section 4, line 577 : "We remind that we fixed the value of veff using multi-angular polarized measurements of OSIRIS, which leads to choose a weak uncertainty of $v_{eff}$ (15%). However, if no information on veff is available in the measurements, the uncertainty should be higher and thus the errors due to the non-retrieved effective variance. Platnick et al. (2017) obtain 2% and 4% uncertainty for COT and Reff respectively for a standard deviation from veff between 0.05 and 0.2"

Concerning the wind speed, we add line 328 : "…with a fixed ocean wind speed based on NCEP reanalysis of the National Oceanic and Atmospheric Administration (NOAA)"

If the authors make the results useful to the future 3MI mission, the uncertainty (specifically, the standard deviation of a model parameter) of each error source (in particular, the model parameter errors) should represent those of global climatology. Specifically, I feel that the cloud top height and effective variance uncertainties may be a bit too small, so the retrieval RSD might be underestimated (although the general conclusion the little contribution of model parameter uncertainties to retrieval RSD may not change).

The reference to the 3MI instruments concerns the use of OSIRIS data, which is a prototype of 3MI and the methodology rather than the results and analysis of our studies. Obviously, for 3MI, the methodology has to be adapted. For example, the uncertainties on cloud top would be larger, as the cloud top pressure uncertainties from the O2 band are larger (Desmons et al., 2013). For the effective variance, it will be possible to add this parameter in the state vector using a combination of the polarized multi-angular and shortwave infrared measurements in the measurements vector.

We mention the adjustments that have to be made for applications of the method to 3MI measurements in the conclusion section, line 740 to 746 :
"Note that, since information provided by Lidar or polarized measurements was used, the uncertainty for the non-retrieved parameters was chosen to be low. For applications to other cases without these available information, errors would be higher. If the method is applied to 3MI for example, the errors related to the cloud top altitude would be higher as the O2-A band leads to cloud top pressure uncertainties between 40 and 80hPa depending on the cloud types (Desmons et al. 2013). A more complex algorithm could also be used with a measurement vector including O2-Aband radiances and multi-angular polarized radiances to have information on and to add the cloud top altitude and the effective variance (Huazhe et al. 2019) in the state vector."

For these reasons, I suggest the authors clearly state what are the focuses of this paper in the last paragraph of the introduction and reconsider the experimental design of this work through, but not limited to, using numerical experiments, adding the retrieval biases to the current analysis, or other appropriate methods.

Using numerical experiments can also be a good way to assess the uncertainties of the retrieved parameters. However, doing it for different types of clouds, different geometry conditions (solar and view) is a huge work, which is beyond the scope of this paper.
To clarify our objectives, we add a paragraph in the end of the introduction section (line 141-146) :
"The aim of this paper is not to give an exhaustive view of the possible errors concerning optical thickness and effective radius retrievals but to simply introduce a method to derive the different sources of uncertainties from a specific case of data acquired during an airborne campaign. Uncertainties due to error measurements, to non retrieved parameters but also to the assumed forward model are considered. If generalized to several cloudy scenes, the partitioning of the errors can help to understand if and which non-retrieved parameters or forward model need to be optimized in order to reduce the global uncertainties of the retrieved cloud parameters."

We agree that a study based on simulated data can validate our framework by showing that errors on retrieved parameters are included in the uncertainties obtained using the methodology presented.
We add in the conclusion section, line 767 to 769.
"A way to check the consistency of the method and the validity of the uncertainty ranges would be to simulate radiances using Large Eddy Simulation model with realistic cloud physical description, add noise for the errors measurements and derive the cloud parameters and their uncertainties."

Analysis flow

While I appreciate the authors for describing the solid mathematical basis of the analysis procedure, I may have an argument on the use of COT and Reff retrievals obtained based only on measurement noise in the error covariance matrix for the following analysis. This can overfit the retrieval variables to obtain an optimal solution (i.e., J ~ ny) because radiative signal perturbations (here what I mean is any of radiative perturbation induced by atmospheric-cloud properties naturally occur in the real world) originating from other sources of errors are partly explained with the biased retrieval values. If such perturbations are significant (and yes, it is significant particularly for vertical heterogeneity and 3D effects), then the sensitivity of the other error sources to the retrieval quantities are obtained from the sensitivities to the retrieval variables at the biased cloud property conditions. Ideally, numerically generated cloud property fields (such as Large Eddy Simulations) would provide datasets for the evaluations of the retrieval uncertainties based on the authors' flamework. As the true cloud properties are not available in the observed cloud field, it is not possible to address this issue based on the given observations. Therefore, at least, the authors should discuss a potential bias of the retrieval uncertainty evaluations based on this framework.

Indeed, the optimal estimation method gives the state vector that best matches the measurement vector under the assumption of a transfer function to pass from the state vector to the measurement vector. If the transfer function is false or biased, the retrieved parameters will obviously be biased. The uncertainties obtained for the non-retrieved parameters or to the forward model will also be incorrect if the variations predicted by the model about the retrieved (biased) values are different from those about the true value.

We add a sentence to raise this issue in the section 4, line 533 to 538 :
"The parameters retrieved in the first step may be biased, in particular due to the use of a simplified cloud model to connect the state vector to the measurements. The estimation of the uncertainties performed in the second step assumes that the variations predicted by the simplified and the realistic models around the retrieved values (potentially biased) and around the true values are identical. This is correct with a linear forward model but can be a too strong assumption in cloud retrieval regarding the non linearity of the relationship of the radiances in function of cloud parameters. A way to test this assumption would be to use numerical experiments"

And we add in the conclusion section, line 763 to 768 :
"The method was applied to real data, which means that the true cloud parameters are unknown. Consequently, it is not possible to know if real errors on the retrieved parameters are included in the uncertainties given by the method presented here. One reason that can lead to an erroneous assessment is that the estimations of the uncertainties are done around the retrieved values than can be biased. A way to check the consistency of the method and the validity of the uncertainty ranges would be to simulate radiances using Large Eddy Simulation model with realistic cloud physical description, add noise for the errors measurements and derive the cloud parameters and their uncertainties."
"

**Minor comments**

1. Title: Actually, the authors use only two bands for the cloud property retrievals, and therefore, "bispectral" measurements would be more descriptive rather than "multispectral" measurements?
Agree, it was done

2. Line 241, Eq. (7): Why do the authors use a linear scale of the cloud optical thickness for a state vector element, not a logarithmic scale? Although the relation between solar reflectance and COT is linear at very small COT conditions ($t \ll 1$), it is in general quite non-linear over most of the COT range, which makes the retrieval process slow and may degrade the convergence, and also may limit the representativeness of this results to optically thin cases.
We agree that using a logarithmic scale could help the convergence but we do not use it.
We mention in the text, that it could accelerate the convergence, line 306 :

"It can be noted because the relationship between radiances and optical thickness has a logarithmic shape, using *log(COT) i*nstead of COT in the state vector may accelerate the convergence."

3. Section 3.3: I do not find the values of a priori error covariance matrix in the manuscript. Please briefly mention what values are chosen.
It is an omission in the text, we precise the a priori vector and  Sa value, line 309 : "The a priori state vector was set to [10,10µm] and the a priori covariance matrix Sa was set to 108. The latter was chosen very large in order to favor the measurements in the determination of the state vector."

4. Line 338 "$\sigma$alt = 0.16" :: This represents only the cloud properties observed during the CALIOSIRIS campaign.
As explained above, this corresponds to the specific case of the CALIOSIRIS campaign. If the same methodology is followed for another campaign or another experimental setup (for example without LIDAR), the value can differ. To avoid misunderstanding, we add in line 423: "This value is low thanks to the knowledge provided by the Lidar."

5. Line 333 "$\sigma$veff = 0.003" :: How did you get this value? It seems too small. Please site references that support this quantity.
See the answers and modifications made in the general comments response.

6. Lines 372–376: In addition, the collision-coalescence process can provide a larger droplet radius at the lower part of clouds, which are observed from CloudSat. As vertical heterogeneity is important to the cloud property retrieval, the authors may consider a better representation of cloud profiles using a better cloud profile parameterization (e.g., Saito et al., 2019).
We add it in the introduction section, line 114:
"Saito et al. (2019) propose a method to retrieve the vertical profile using Empirical Orthogonal Function (EOF) to reduce the degrees of freedom of the droplet size profile"

And in section 3.3.3 line 463:
**Depending on the maturity of the cloud**, turbulent and evaporation processes can reduce the size of droplets at the top of the cloud **and collision and coalescence process can increase the size of the droplets in the lower part of the clouds as observed by Doppler Radar  (Kollias et al., 2011). The profile used in this study aims to represent the case of droplet size reduction at the top of the cloud but other and more sophisticated and representative profiles can be used (e.g. Saito et al., 2019).**

7. Lines 416–420: As the authors assume the flat cloud top, which reduces some of cloud 3D effects such as illuminating and shadowing effects, I had an impression that the authors may focus on lateral photon transport effects here. If so, it would be better to rephrase 3D with lateral photon transport or state "3D" regarded as the lateral photon Transport.
In the paragraph describing the 3D radiative transfer simulations, we refer to 3D and 1D radiative transfer, so we decided to keep the term "3D" but add a sentence to express that differences between the two are related to the heterogeneity along the lines of sight and lateral photon transport, line 519 :
"The differences are thus mainly due to the lateral photon transport which tends to smooth the radiances fields compared to their 1D counterpart (Davis et al. 1997) and to the cloud heterogeneity along the line of sight (e.g. Fauchez et al., 2018)."

8. Figure 4: The readers cannot recognize if there are optimal/non-optimal solutions from these plots. If it is non-optimal, a set of cloud retrievals may not adequately explain the measured signals. I suggest the authors add the cost function distributions in addition to these two plots.
We add the normalized cost function  and the convergence type in Figure 4 and Figure 10. Values less or equal to one indicate a convergence of type 1 represented in green in Figure 4 and Figure 10. If convergence of Type 1 does not occur, the iteration can stop with convergence of Type 2 when the

difference of the state vector between two successive step are less than $n_X$

We add in the comment of Figure 4, line 542 to 550: "Figure 4c presents the normalized cost function, which is less or equal to one when the retrieval successfully converges according to Eq. 6 (convergence of Type 1). In case of multi-angular measurements, the normalized cost function is often above one meaning that the simulated radiances do not fit the measurements while considering the measurements error covariance only. This comes from the attempt to fit the measured radiances from all the available viewing directions with a too simple forward model far from reality. The retrieval stops thus mainly according to Eq. 7 (convergence of Type 2) indicating that the state vector remains almost constant between two successive iterations. When neither Eq. 6 or Eq. 7 are achieved the retrieval fails. For the whole scene, failed retrievals account for 3.3% of the pixels. The failure may be associated with pairs of radiances outside the LUT that can occur for several reasons well documented in Cho et al. (2015)."

And in the comment of Figure 10, line 677 :
**"A normalized cost function value (Figure 10c) less or equal to one** is not necessarily an indication of an accurate retrieval, but only that a fit occurred."

9. Lines 483–484 "3D effects due to solar illumination do not appear in the retrieved cloud properties" :: This is an obvious statement as the cloud top is assumed to be flat, which removes the shadowing and illuminating effects. Please state that this error evaluation focuses solely on the lateral photon transport effect.
Even if illumination and shadowing effects are also present to a lesser extent for flat cloud top (Varnai et Davies 1999), we remind that we assume a flat cloud top that minimizes the solar and shadowing effects, line 612 : "However, in this work, we are dealing with flat cloud tops that induce weaker 3D effects than bumpy cloud tops (Várnai and Davies, 1999)**"**

10. Lines 539–540: The cost function divided by the number of measurement signals (J/ny) is a comparable quantity among mono-angular- and multi-angular-based retrievals.
We agree that it is comparable but we chose to compare RSD to be consistent with the rest of the paper. We add that we can also compare this quantity, section section 5, line 693:
"To compare the uncertainties of the two retrievals , we use the relative standard deviation (RSD) **to be consistent with the previous results**"

11. Lines 559–560" Why are these uncertainties reduced for multi-angular cases?

Multi-angular measurements provide more information to constraint the state vector, especially in the cloud bow regions, leading to a reduction of the RSD. This was already reported in line 669:
"Clearly, the multi-angular measurements contain more information and allow to resolve the problem encountered with the mono-angular bispectral method which is also clear in the reduction of the failed convergences from 7.6% to 3.3%."

12. Line 610: If the authors state that "the uncertainties related to the measurement errors is implementable in an operational algorithm," the uncertainty evaluations should be based on global climatology of cloud, surface, and atmospheric properties. A limited case (i.e., a granule of a cloud scene) may not be adequate to state so.
In this sentence, we refer to the account of the measurement errors that are included in the PDF of the measurement vector through their standard deviation. For the other types of errors that are currently hardly implementable in an operational algorithm due to computational cost reasons, using climatology can be a good solution. We complete the sentence about these others sources of uncertainties, section 6, line 775:
"The second step that consists of computing the uncertainties resulting from the non-retrieved parameters and from the forward model is more computationally expensive but could also be included. The uncertainties related to the non-retrieved parameters, in addition to the one related to measurement errors, have already been implemented since Collection 5 in MODIS operational algorithm through the computation of covariance matrix where Jacobian are derived from look-up table and was completed

for Collection 6 (Platnick et al. 2017). Concerning the forward model errors, the method cannot be implemented as in this work in an operational algorithm because of the prohibitive computation time. A climatology based on several cases studies, depending of the type of clouds, land or ocean surface flag for example could be used in order to obtain a distribution of the errors according to the scene characteristics"

13. Typo and grammatical errors: Please proofread the main body of the manuscript again. I have found several grammatical errors, e.g., Lines 557, 571, and possibly more.
We apologize for these typos and grammatical errors and have again done a careful proofreading, hoping to have almost removed the typo and grammatical errors.

Reference
Saito, M., P. Yang, Y. Hu, X. Liu, N. Loeb, W. L. Smith Jr., and P. Minnis, (2019) An efficient method for microphysical property retrievals in vertically inhomogeneous marine water clouds using MODIS–CloudSat measurements, J. Geophys. Res. Atmos., 124, 2174-2193.

**Reply To RC2:**

We are deeply grateful to the Reviewer for the time spent to carefully read our paper and for his constructive and relevant comments. They helped us to improve our article considerably. Please find hereafter our point-by-point responses to the comments and suggested corrections. Comments are in black, our responses in blue and the text modifications in green (in bold when only one part of the sentence was modified).
Note that the indicated line numbers correspond to the line numbers of the "track changes" version.

RC2:

Review of 'Cloud optical properties retrieval and associated uncertainties using multi-angular and multi-spectral measurements of the airborne radiometer OSIRIS'
By Christian Matar et al.

General comments
This paper describes an approach for the retrieval of cloud optical thickness and droplet effective radius from multi-angular and multi-spectral satellite measurements. In addition, estimates of the retrieval errors due to different error sources are presented. The main novelty is an attempt to estimate retrieval errors caused by deviations from vertically homogeneous profile and independent pixel assumptions. The topic is important and suitable for this journal. Overall the retrieval setup appears to be sound and the results are plausible. However, I do have some general concerns and questions that need to be addressed before this paper can be published.
• Only one case is studied, which limits the validity of the results. Retrieval errors, in particular those caused by deviations from homogeneous cloud and independent pixel assumptions, will depend on the scene. While discussing more cases may be outside the scope of the paper, these limitations at least need to be mentioned.

This point was also raised by the first reviewer. To clarify our objectives, we add a paragraph in the end of the introduction section :
"The aim of this paper is not to give an exhaustive view of the possible errors concerning optical thickness and effective radius retrievals but to simply introduce a method to derive the different sources of uncertainties from a specific case of data acquired during an airborne campaign. Uncertainties due to error measurements, to non retrieved parameters but also to the assumed forward model are considered. If generalized to several cloudy scenes, the partitioning of the errors can help to understand if and which non-retrieved parameters or forward model need to be optimized in order to reduce the global uncertainties of the retrieved cloud parameters."

• The paper gives a false impression of the current cloud optical and microphysical property

algorithms. Firstly, most of these do consider measurement errors and produce retrieval error estimates (e.g., Platnick et al., 2017), unlike what is written in the abstract. Secondly, for many of these algorithms the total retrieval error has been separated into contributions from individual error sources (e.g., Walther and Heidinger, 2012), unlike what is written on page 10.

We apologize for these major omissions and thank the reviewer for giving us the references. We correct the text accordingly and add the corresponding citations.

In the abstract, we delete : "...and without considering measurement errors and the choice of ancillary data"

In the introduction, we replace line 120 "But most importantly it also lacks the ability to assess the uncertainties on the retrieved properties." by "In addition, until recently, the difficulty was to assess the uncertainties of the retrieved cloud properties. Platnick et al. (2017) succeeded to derive the total uncertainties of COT and Reff and to decompose the contribution of uncertainties from measurement errors and from several non retrieved parameters, using covariance matrix and Jacobian computations from LUT."
"

In the introduction, line 137, we complete (in bold) the following sentence : "Therefore, it introduces the probability distribution of solutions where the retrieved parameter being the most probable, with an ability to extract **separately** uncertainties on the retrieved parameters **(Whalter et Heidinger, 2012)**."

We correct the false assertion page 10, we replace section 3-3 line 361  "The contribution of each type of error was not separated. To highlight their magnitude and better understand the sources of errors in cloud retrieval algorithms, we separate the contributions of each type of error"
by "Further, Walther and Heidinger (2012) use the optimal estimation framework to separate the contribution of measurement errors and several non retrieved parameters. In our work, a similar framework was used to separate the contribution of each type of uncertainties including also the forward model uncertainties to better quantify and understand the limitation of using simplify forward model in such cloud retrieval algorithm."

In the conclusion section, we add line 777:
"The uncertainties related to the non-retrieved parameters, in addition to the one related to measurement errors, have already been implemented since Collection 5 in MODIS operational algorithm through the computation of covariance matrix where Jacobian are derived from look-up table and was completed for Collection 6 (Platnick et al. 2017)."

• It is not clear why polarization measurements have not been included in the retrieval. It looks like these could, amongst others, further constrain the width of the size distribution, which has instead been fixed prior to the retrievals.

The OSIRIS instrument used for this study is under development in the Laboratoire d'Optique

Atmosphérique. At the time of the CALIOSIRIS mission in 2014, the polarized measurements had calibration and straight light issues. The measurements were consequently not usable to do a quantitative retrieval.

We precise this issue, in section 2.2, line 218 : "At the time of the CALIOSIRIS campaign in 2014, the polarized channels presented calibration and stray light issues, which make use of the polarized measurements difficult for quantitative retrievals. In addition, the images from the two sensors were not well co-localized. Consequently, for this work, we use the two channels of the SWIR matrix, one almost non absorbing (1240 nm) and one absorbing (2200 nm) to have information on optical thickness and effective radius respectively."

Even if the polarized radiances were not used for the retrieval, we used an averaged value to determine the effective variance according to the number of the supernumerary bows. Figure 1 represents averaged polarized radiance measurements obtained with OSIRIS and simulated radiances, with an effective variance of 0.02, that has been selected.

[Figure]

Figure 1 (not included in the paper) : Averaged polarized radiances measured by OSIRIS for a transect in the middle of the central image of CALIOSIRIS scene and simulated polarized radiances with an effective variance of water droplet distribution equal to 0.02 (in blue), as a function of the scattering angles.

We complete the sentence, line 428: "...we fixed a value of 0.02 based on the **number** of supernumerary bows in the polarized radiances **(not shown)**.

and line 430: As the value of $V_{eff}$ was fixed using the polarization measurements of OSIRIS, this uncertainty is weak and is not representative of all clouds"

We add also in the analysis of the non retrieved parameters, line 577 : "We remind that we fixed the value of veff using multi-angular polarized measurements of OSIRIS, which leads to choose a weak uncertainty of veff (15%). However, if no information on veff is available in the measurements, the uncertainty should be higher and thus the errors due to the non-retrieved effective variance. Platnick et al. (2017) obtain 2% and 4% uncertainty for COT and Reff respectively for veff between 0.05 and 0.2."

• There are many textual mistakes and inaccuracies, of which only some examples have been

included in this review.

We deeply thank the reviewer for his careful reading, the time spent to read the paper and apologize for the different mistakes. We correct it at our best.

Specific comments

Title: The paper only deals with liquid clouds. This must be included in the title.

Done

Abstract, L13: Clouds are really characterized by more properties than these two, e.g. the height of the top and base.

We agree, we add "...and additionally by geometric properties when specific information is available"

Abstract, L16: As mentioned above, measurement errors are considered in most retrieval algorithms.

The sentence was deleted, see above

Abstract, L26: I suggest not to refer to 'traditional' bi-spectral retrievals as 'MODIS-like methods'

since these methods are not in any way specific to MODIS.

Done in the abstract and afterwards: "MODIS-like" is replaced by "mono-angular bispectral method"

P1, L36: The statement is still true so why not cite the latest IPCC report here?

Because the paper was written before its release. We update the reference

P2, L40: Measurements of emitted radiation are also important for cloud property retrievals.

We add it

P2, L50: Suggest to cite the more recent Platnick et al. (2017).

Done here and afterwards

P2, L51: The bi-spectral method is applied to many instruments, not 'in particular' to MODIS.

We replace "in particular" by "for example"

P3, L93: What is 'roughening of the radiative field'?

We complete the statement by adding "...by increasing or decreasing radiances compared to the prediction of the plane-parallel homogeneous clouds."

P4, L110/111: LUT-based retrieval algorithms are very well suited to produce retrieval error estimates. Please remove this statement.

We delete it and add "Platnick et al. (2017) succeeded to derive the total uncertainties of COT and Reff and to decompose the contribution of uncertainties from measurement errors and from several non retrieved parameters, using covariance matrix and Jacobian computations from LUT."

P4, L125: 'model': do you mean 'state'?

That's right, we modify it.

P5, Fig. 1: The figure and caption can be improved. Lines cannot be distinguished very well. What are 'optical matrices'? Spectral response functions is probably a more common term. And in what units are these plotted / how have they been scaled? The y-axis only says 'Transmittance'

The caption was completed: "Spectral wavelengths of VIS-NIR (left) and SWIR (right) **spectral response function of OSIRIS** optical **channels** without unit and normalized to unity.

The dashed line corresponds to a typical atmospheric transmittance **in %**. The red-colored channels are used in this study (1240 and 2200 nm)."

P5, L163-165: Can you expand a bit on the viewing angles? Here numbers of 19 and 20 angles are mentioned, later it is 13 angles. How many angles were actually used for the retrievals? Can you also comment on how well the pixels for the different viewing angles are aligned? And if they are aligned at ground level, isn't there a spatial mismatch at cloud altitude?

19 or 20 angles corresponds to ground level colocalization.

We explain now in section 2.2, line 223 how the successive images are colocalized and mention that only 13 directions are used because of the cloud altitude : "In order to use the multi-angular capability of OSIRIS, successive images have to be colocalized. After subtracting the average of similar successive images to remove the angular effects, the colocalization is achieved by minimizing the root mean square difference of the radiances between each pair of successive images for different translations along the line and the column in the second image. The reference image is the central one of the sequence. Images with translations beyond its dimensions are ignored. Multi-angular radiances at the cloud level correspond in our case to 9 to 13 directions."

P6, L175: What is 'LIDAR-LNG'? And what is 'the vertical profile' in Fig. 2b? There is a whole series of vertical profiles.

We add a sentence and a reference to introduce the LIDAR-LNG: "The LNG (lidar aerosols nouvelle generation, Bruneau et al. 2015), a high spectral resolution airborne Lidar at 355nm, was also onboard the Falcon-20 aircraft along with OSIRIS during the airborne campaign."

And correct "the vertical profile**s of the backscattered signal**" in the text and in the caption of Figure 2

P6, L179: There are some 'intense white' but also quite dark parts in the reflectance image. Please explain what you mean.

There is not really an intense white signal but looking carefully, we can see the decomposition of the light between the scattering contour lines 140° and 150°. We correct the sentence and write, line 205 :

"The clouds backscatter total solar radiation **more intensely** in the cloudbow regions near 140°. The position of the cloudbow peak depends on the wavelength, resulting in the **decomposition of the light slightly visible between the 140° and 150° scattering angle contours.**"

P6, L182: Where do you see a 'white arc' in Fig. 2d? I only see an arc with rainbow colors.

We modify:

On the polarized image (Figure 2d), we observe a strongest directional signature of the signal, characteristic of scattering by spherical droplets s**howing a cloud bow  clearly visible between about 140° and 150°**.

P6, L185: Could you perhaps indicate in the figure where the reflectance is (or may be) affected by sun glint?

We indicate in the text, line 213 :

"Since the solar zenith angle is 59°, the specular direction corresponds to a scattering angle of 62° in the solar plane (not visible in Figure 2) but the ocean wind enlarges the sun glint area, which enhances the radiances between the 70° and 80° scattering iso-contours."

P7, Fig. 2 and caption: Please use UTC (not local time) throughout to avoid confusion. What is the background in Fig. 2a? Please add explanation of red arrow in Fig. 2a and red rectangle in Fig. 2b in the caption and not (only) in the text. What is the half-visible text on top of that image? What are all the colors in the legend (most of which do not appear in the figure)? What is the blue bar on the left hand side of Fig. 2c? I don't think that 490-670-865 nm is a 'true-color' RGB composite. From which viewing angle are these radiances? The contours are not concentric.

The text at the top of the Fig2-a is only the filename of the image. We delete it. The blue bars on the left hand side of Fig 2C are related to the motion of the airborne between the different filters, which are on a rotating wheel. One OSIRIS image corresponds to several viewing angles represented by the iso-contours of the scattering angles. We add text on it in the legen. In addition, we modify the caption according to the comments (modifications in bold):

"Figure 2: Studied case on 24 October 2014 **at 10:02 UTC** (11:02 local time): (a) **In blue,** airplane trajectories for this day **above a MODIS/AQUA true color image**. **The red arrow corresponds to the studied segment** (b) Quicklook **of the backscattered signal** provided by the LIDAR-LNG around the observed scene. **The red rectangle corresponds to the scene studied** (c) OSIRIS  RGB composite image, obtained from the total radiances at channels 490, 670, and 865 nm. **The blue bars on the left hand side of the images are due to the motion of the airborne between the capture of the image acquisitions of the different filters**
(d) OSIRIS  RGB composite **image**, obtained from the polarized radiances at channels 490, 670, and 865 nm. The white  **iso-**contours in (c) and (d) represent the scattering iso-angles in a 10° step."

We also add in the text, the range of zenith angle, line 201:
"One OSIRIS image corresponds to several viewing angles. The zenith angle range from about 0° in the center of the image to 55° in the corner of the image."

P8, L205: It is stated that vector y has dimension n_y but in Eq. (8) it has dimension 2*n_y.
The dimension of the measurement vector is well n_y but the dimension in Eq. 8 (now eq. 9) should be n_theta. We correct it.
P8, L218: Please state explicitly how x_a and S_a are defined ('large' is too vague).We add section 3.2, line 309 : "The a priori state vector was set to $[10,10\mu m]$ and the a priori covariance matrix $S_a$ was to $10^8$. The latter was chosen very large in order to favor the measurements in the determination of the state vector (no a priori constraint)."

P8, L225: Suggest to refer to page 10 where the Jacobian is explained.We refer now to Eq. 10
P9, L250-251: Is this justified? From Fig. 1 it looks like there is considerable absorption, in

particular in the 2.2 micron channel, which needs to be accounted for.

We agree with the reviewer that for a proper and operational algorithm, atmospheric corrections should be done. Not accounting for atmospheric absorption in the short-wave infrared bands, lead to a smaller optical thickness and a larger effective radius.

We add, line 322: "As it is not completely true, the retrieved cloud optical thickness will be slightly underestimated and the effective radius slightly overestimated."

P9, L255: What are these 'measurements'?

It was not measurements but the NCEP National Centers for Environmental Prediction) atmospheric reanalysis. We correct :

"…with a fixed ocean wind speed based on  NCEP reanalysis of the National Oceanic and Atmospheric Administration (NOAA)"

P9, L258: What is 'independent column approximation'? Is it the same as the 'independent pixel approximation' introduced on page 2? If so, please use consistent terminology.

We correct and use independent pixel approximation, which is the usual term.

P9, L264-268: How do you calculate the Jacobian matrix?

We add, line 340 : "The Jacobian Matrix is computed using finite differences."

P10, L274: The off-diagonal terms are non-zero so it is confusing that they have been written here as zeros. Can you explain?

We assume that the uncertainties for the two terms are independent. Consequently, the off-diagonals matrix terms are null. We add line 353 : "In this formulation, we have assumed that the two terms of the state vector are independent, thus the off-diagonal terms of Sx are assumed to be zero."

P10, L276: Do you mean Eq. (7) instead of Eq. (10)?

Yes, corrected

P10, L287: As mentioned in the general comments, there have been other studies where the types of error were separated.

It is corrected with the insertion of the reference, Walther and Heidinger (2012). See general comments answers

P11, L300: There are other ways of calibration, e.g. vicarious.

We add : "...or can be vicarious calibration (e.g. Hagolle et al. 1999) by using natural or artificial sites on the surface of the Earth"

Hagolle O, Goloub, P, Deschamps, P-Y., Cosnefroy H., Briottet X., Bailleul T., Nicolas J-M, Parol F., Lafrance B., and Herman M., Results of POLDER in-flight calibration, IEEE Transactions on Geoscience and Remote Sensing, vol. 37, no. 3, doi: 10.1109/36.763266.

P11, L301: I would argue that calibration usually addresses systematic (not random) errors in the measurements.

Once the calibration process has been done and the correction coefficient applied, uncertainties remain. We correct the sentence : "The uncertainty of the measurements

 **remaining after the calibration process** are assumed random, uncorrelated between channels…"

P11, L304: According to Eq. (8) the dimension of the measurement vector is 2 n_y x 2 n_y (?)
Eq. 8 was corrected.
P11, L306: Do you have a motivation for taking a fixed 5% measurement error?
This error was estimated by the engineers working in the development of OSIRIS

P12, L341: This value of v_eff is quite small. Could you add some explanation on how it is determined? Wouldn't it be possible to include its determination in the overall retrieval (by including the polarization measurements in the observation vector)?
As explained in the general comment, the Veff value was chosen using the multi-angular polarized measurements of OSIRIS. They were not well calibrated but allow to see the number of supernumerary bows. As the help of polarized measurements allows to fix Veff value to 0.02, an uncertainty of 15% was added.
We add, line 428 : "..we fixed a value of 0.02 based on the **number** of supernumerary bows in the polarized radiances (**not shown**)."
and in the analysis of the non retrieved parameters, section 4, line 577 : "We remind that we fixed the value of veff using multi-angular polarized measurements of OSIRIS, which leads to choose a weak uncertainty of veff (15%). However, if no information on veff is available in the measurements, the uncertainty should be higher and thus the errors due to the non-retrieved effective variance. Platnick et al. (2017) obtain 2% and 4% uncertainty for COT and Reff respectively for a standard deviation from veff between 0.05 and 0.2."

P12, L336-343: In this case the cloud top height (and thickness) and effective variance can be determined very accurately. However, in 'real life' uncertainties will be much larger (e.g., if you have no lidar available). Wouldn't it therefore be better to work with larger uncertainties so that the resulting error estimates become more representative?
We agree, but as we answer to the first reviewer, our motivation was to present a method that can be applied to 3MI or any other sensors and not to give an overview of the range for each type of error.
The algorithm was implemented according to the information available during the CALIOSIRIS campaign. The LIDAR measurements was one of them. We add several paragraphs in the introduction and conclusion sections to clarify our objectives.
More specifically, concerning the uncertainties of the non-retrieved parameters we add line 418 :
"The values and the uncertainties of the fixed parameters are chosen according to the experiment setup of the campaign."
And we add in the conclusion section :
" Note that, since information provided by Lidar or polarized measurements was used, the uncertainty for the non-retrieved parameters was chosen to be low. For applications to other cases without these available information, errors would be higher. If the method is applied to 3MI for example, the errors related to the cloud top altitude would be higher as the O2-A band leads to cloud top pressure uncertainties between 40 and 80hPa depending on the cloud types

(Desmons et al. 2013). A more complex algorithm could also be used with a measurement vector including O2-Aband radiances and multi-angular polarized radiances to have information on and to add the cloud top altitude and the effective variance (Huazhe et al. 2019) in the state vector."

P13, L355-358: In the OE framework errors are assumed to be Gaussian and error estimates reflect 1-sigma of the Gaussian distributions. Can you comment on the Gaussian nature of the forward model related errors? Is it plausible to use the difference between two configurations as 1-sigma of the uncertainty, or would these configurations rather reflect two extremes?

Indeed, the OE method is based on the assumption of Gaussian distribution. If it is not the case and if biases related to the forward model exist, they are included in the Gaussian distribution. This leads off course to an overestimation of the uncertainties.

In the case studied here, the assumption of a Gaussian distribution is not too bad. Indeed, if we look at the Figure 9 of the paper, the difference between 1D and 3D radiances is not so far from a Gaussian distribution. A. This happens because the cloud is flat and the spatial resolution high (about 20m), so the sub-pixel bias can be neglected. However, at a larger resolution, this bias can not be neglected and will increase the uncertainties.

We add in section 3.3.3, line 448: "The simplified model used for the retrieval can lead to biased retrieved parameters. In this case, the bias will be included in the Gaussian PDF width, resulting in an overestimation of the uncertainties."

P13, L358: What is the square of a matrix?

We correct and use the transpose matrix

P13, L363: To estimate retrieval errors due to deviations from the assumption of vertical homogeneity, a specific alternative cloud model is outlined in detail. However, it should be realized

that this is just one possibility. For example, real profiles have a varying degree of sub-adiabaticity, which is not considered here. What would be the effect on the uncertainty estimates?

We agree and this point was also raised by the first reviewer. We write, line 466:

**Depending on the maturity of the cloud**, turbulent and evaporation processes can reduce the size of droplets at the top of the cloud **and collision and coalescence process can increase the size of the droplets in the lower part of the clouds as observed by Doppler Radar (Kollias et al., 2011). The profile used in this study aims to represent the case of droplet size reduction at the top of the cloud but other and more sophisticated and representative profiles can be used (e.g. Saito et al., 2019) "**

P14, Fig. 3: The cloud is placed between 5 and 6 km. Where do these numbers come from? They are not the same as on page 12. Please include the settings (top and bottom height, cloud optical thickness, maximum effective radius, …) for this particular figure in the caption.

The cloud is between 5 and 6 km. We correct the value page 12.

We add the cloud properties in the caption of the Figure 3 :

The cloud is between 5 and 6km. The maximum extinction coefficient and effective radius are 6.6 km-1 and 12 μm respectively and the altitude zmax is 5.85km

P15, L396-398: Is it correct to determine the maximum effective radius such that the average effective radius of the heterogeneous and homogeneous profiles are the same? Shouldn't R_eff be weighted with extinction? Or, alternatively, a requirement to arrive at the same liquid water path for both profiles seems better justified.

Several solutions can be used, and we chose $Reff_{max}$ in order to have the mean Reff of the heterogeneous vertical profile equal to the Reff of the homogeneous cloud. We add in the text that other options can be possible, we add line 490. "Several options are possible for these values."

P15, L407: Only the IPA seems to be addressed here. What about the PPH assumption?

To clarify it, we correct line 496, by replacing "IPA" by "PPH" line and add a sentence to describe the PPH effects (next comment) and another one to explain the choice of studying only the IPA errors, line 509: "At the high spatial resolution of OSIRIS (less than 50 m at the cloud level), it was shown from airborne data that the dominating effect is related to the IPA error (Zinner et Mayer, 2006). In the following, we consider thus only this error and assume that the pixel is homogeneous at the measurement scale."

P15, L409-411: It seems to be stated that the PPH assumption includes the IPA, whereas in earlier parts of the manuscript they were introduced as different things (which I think they are).

The Plane Parallel Homogeneous (PPH) cloud assumption includes the PPH bias due to subpixel variability of the cloud and the IPA errors due to transport of horizontal photons not accounted for with the PPH assumption. To clarify if we add, line 508 "This PPH assumption includes errors known as the PPH bias due to the subpixel variations of the cloud and errors related to the photon horizontal transport between columns (IPA error)"

P16, Fig. 4: Does the scene contain clear-sky pixels? If so, how are they reflected in the COT and Reff maps? Are there any failed retrievals? If so, how are they reflected in the maps?

The studied case is a stratocumulus, with thin optical thickness but no clear sky pixel.

We add line 541 "Some values of COT are very small but no clear sky pixel is present."

P16, caption Fig. 4: Is the date a typo or is this really a different case from the one introduced in Fig. 2?

It was a mistake and was corrected

P16, L436: A figure with COT and Reff uncertainties as functions of COT and Reff would be very instructive to illustrate this.

We add COT and Reff uncertainties as functions of COT and Reff values in Figure 5 (panels c et d) and add comments in section 4, line 541 :

"These uncertainties are plotted according to their respective values in panels (c) and (d). RSD COT (mes) increases with the magnitude of the retrieved COT, as RSD Reff (mes) with Reff for values until 15 µm. The uncertainties due to measurement errors are low, especially for optical thickness (less than 5%)."

.

[Figure]

**Figure 5 (revised version):** Uncertainties (RSD (%)) of COT (a) and $R_{eff}$ (b) originating from the measurement errors for the case study of CALIOSIRIS. **COT uncertainties in function of COT (c) . Reff uncertainties in function of Reff (d).**

P16, L435: The uncertainties in COT appear to be very low. Is a retrieval error of 0.5% realistic? For thin clouds COT depends approximately linearly on the reflectance. How can a reflectance measurement uncertainty of 5% result in an order of magnitude lower uncertainty in COT? Is this thanks to the combined information from different viewing angles. But, if that's the case, isn't the assumption of uncorrelated errors between the measurements from these different angles much too optimistic?

Two reasons explain this low value : the quasi-linearity and the steep slope of the radiance as a function of COT and Reff and the multiangular measurements. Indeed, this is one main advantage of the optimal estimation method, which allows the use of several measurements per cloudy pixel to obtain the best estimation of an unique COT value. For the same pixel, each additional information reduces the uncertainty on the retrieved parameters in the presence of the same 5% random noise in the measurement.

We mention, line 557 "...the steep slope…" of the radiances in function of COT.

The advantage of using multiangular measurements is mentioned several times in the paper. For example, line 699:

"The multi-angular approach leads indeed to more information available for each cloudy pixel and each additional information reduces the uncertainty on the retrieved parameters in the presence of the same 5% random noise in the measurements."

Concerning the correlation between the measurements from different views, we agree that it could exist but they are currently not characterized. We add a mention of these possible correlations in the conclusion section, line 735: "Since they are not characterized, the correlations between the measurements issued from different viewing angles are not considered in our retrieval, but they could increase these values."

P17, 459-460: Could this estimate be too optimistic? In case of broken clouds, sun glint can have a relatively much higher impact on the measured reflectance, which would not be captured here.

As we wrote above, the cloudy scene studied is completely overcast but, of course in case of broken clouds the errors due to the ocean wind speed uncertainties will be higher, even if the multiangular measurements can mitigate the effects of these errors.

We add, line 588 : "In case of broken clouds, the errors resulting from the ocean wind speed uncertainty would be larger."

P18, L474: For COT it seems to be rather something like 8 %.

Right, we separate the two : "... to 8% and 20% respectively."

P19, L481-484: This is a firm statement, for which no evidence is provided.

We change the verb to be more cautious and add also the fact that the cloud is flat, which mitigates illumination and shadowing effects: "**However, in this work, we are dealing with flat cloud tops that induce weaker 3D effects than bumpy cloud tops (Varnai et Davies, 1999). In addition,** with multi-angular measurements, the same cloudy pixel is viewed under different viewing angles, which **may** tend to mitigate the influence of illumination and shadowing effects."

P19, L485-486: There are no sub-pixel measurements, and sub-pixel cloud variability is not represented in this work. Again, this is a statement without proof. A PPH error estimate should be added to the retrieval setup, so the PPH effect can be quantified.

That is right. As mentioned above, we assume that the PPH bias is negligible and do not include sub-pixel spatial variability in the simulations and then we are not able to estimate it.

We delete line 616-618 and add line 606 : "We remind here that, given the high spatial resolution of OSIRIS measurements, we consider the PPH bias as negligible and do not account for the sub-pixel variability of cloud properties in the 3D radiative transfer simulation."

P20, Fig. 8: How is radiance defined here? Is it the sun-normalized radiance or true reflectance?

It is reflectance. We modify the caption

From which of the 13 viewing angles are these measurements taken?

It corresponds to the central image of the sequence used for multiangular retrievals but one OSIRIS image corresponds to several viewing angles. It is now indidated section 2.2

Fig. 9: Nice figure, illustrating the different response of thin and thicker cloud portions to 1D versus

3D radiative transfer.

Thank you

P21, L508-509: Is the nearest-nadir view used for the mono-angular retrievals?

The mono-angular retrieval is done according to the geometry of the central images of the series used for the multiangular retrieval. Zenithal angles range from 0° to 55° and azimuthal angle from 0 to 360 (See figure below).

[Figure]

Figure 2 (not included on the paper) : Zenith (a) and azimuthal angle (b) of the central OSIRIS image. Histograms of the zenith (c) and azimuth angle (d).

The zenith angle range is now mentioned in line 201 : "One OSIRIS image corresponds to several viewing angles. The zenith angle ranges from about 0° in the center of the image to 55° in the corner of the image."

P21, Fig. 10: I am shocked by the enormous differences between the mono- and multi-angular retrievals. Ok, for the cloud bow geometries it is well known that mono-angular retrievals do not work. However, for other geometries the mono-angular retrieval should give a reasonable solution, in particular for a reasonably 'well-behaved' cloud field as studied here. This asks for further clarification. Can you also include a scatter-density plot comparing COT and Reff from the two retrievals on a pixel basis?

The differences were enlarged because the color scales were not the same. We modify it in Figure 4 and 10 to have the same color scales in the revised version of Figure 4 and Figure 10. The retrieval looks more consistent except in the cloud bow region and in some cloud parts that may be particularly heterogeneous. The comparisons of the two figures clearly show, as it is already mentioned in the text, higher values of optical thickness and effective radius in case of

mono angular retrievals. Figure 3 presents the scatter plots of the retrieved parameters using multi-angular and mono-angular and confirms this behavior. The correlation between the two optical thickness is good with higher value obtained with mono-angular retrieval. For effective radius, the values are more dispersed but we can still see a relationship between the two effective radii. In the paper, we choose to add in Figure 10 (panels e and f) represented below, the spatial difference between the two retrievals.

[Figure]

Figure 3 (not included in the paper) : Scatter plots of the COT (left) and effective radius (right) multi-angular and mono-angular retrieval

[Figure]

Figure 10 (revised version): COT (a) and $R_{eff}$ (b) retrieved using mono-angular bispectral method for the CALIOSIRIS liquid cloud case study on 30 June 2014 at 11:02 (local time). **Pixels associated to failed retrievals are represented by white pixels. (c) Normalized cost function. Convergence type (Eq. 6 for Type 1 and Eq. 7 for Type 2) and failed retrieval (d). Differences between mono-angular and multi-angular retrieval for retrieved optical thickness (e) and for retrieved effective radius (f).**

Concerning the comparisons of the two retrievals, we add comments in section 5, line 651-655: **The results are presented in Figure 10.** The retrieved COT over the whole field varies between 1 and 12 with a mean value equals to 3.44 **Comparing to multi-angular measurements (mean COT of 2.13), the retrieved COT values tend to be higher**. The range of retrieved Reff has a mean value of 15.65 µm, **compared to 8.76 µm for multi-angular retrieval. Mono-angular retrieval** is particularly affected by the high value of Reff retrieved around the scattering angles 130-140° where the sensitivity of 2200 nm radiances to the water droplet size is known to be small."

P22, L530-531: Apparently both retrievals fail to converge in some cases. But there do not

seem to

be missing values in Figs. 4 and 10. How can that be explained? What output does the algorithm give

in case of no convergence? Are these cases included in the statistics? Are statistics based on a common set of mono- and multi-angular successful retrievals?

The convergence tests used are presented in Eq. 6 and Eq. 7. There is a convergence failure when neither the inequality of Eq. 6 nor that of Eq. 7 is reached after 15 iterations. We add this information in section 2

Line 288: "The iterative process stops when the simulation fits the measurement **(Eq. (5)), named convergence of Type 1** or when the iteration converges **(Eq. (7)) named convergence of Type 2**. **The left side of Eq. (6) represents the normalized** cost function without taking into account the a priori negligible contribution. **When the cost function is smaller$n_y$ than or the normalized cost function (J/$n_y$) less or equal to one,** the iterations stop."

And line 298: "When neither the inequality of Eq. 6 nor  the inequality of Eq. 7 is reached after 15 iterations, the retrieval is considered as failed."

We add the normalized cost function  as panel (c) in the revised version of Figure 4 and 10 and the convergence type and the failed convergence in panel (d) . In the initial version of the paper, the failed convergence were represented by dark blue color. In the revised version, we replace the dark blue color by white color to more clearly show the retrieval fails.

[Figure]

Figure 4 (revised version) : COT (a) and R_eff (b) retrieved **using multi-angular bispectral method** from a liquid cloud case observed during the CALIOSIRIS airborne campaign on 24 october 2014 at 11:02 (local time). **Pixels associated to failed retrievals are represented by white pixels. (c) Normalized cost function. (d) Convergence type (Eq. 6 for Type 1 and Eq. 7 for Type 2) and failed retrieval.**

As comment, we add line 542:

"Figure 4c presents the normalized cost function, which is less or equal to one when the retrieval successfully converges according to Eq. 6 (convergence of Type 1). In case of multi-angular measurements, the normalized cost function is often above one meaning that the simulated radiances do not fit the measurements while considering the measurements error covariance only. This comes from the attempt to fit the measured radiances from all the available viewing directions with a too simple forward model far from reality. The retrieval stops thus mainly according to Eq. 7 (convergence of Type 2) indicating that the state vector remains almost constant between two successive iterations. When neither Eq. 6 or Eq. 7 are achieved the retrieval fails. For the whole scene, failed retrievals account for 3.3% of the pixels. The failure may be associated with pairs of radiances outside the LUT that can occur for several reasons well documented in Cho et al. (2015)."

Concerning Figure 10, we add 655: "This area corresponds also to a more important number of failed retrieval" and line 677 "**A normalized cost function value (Figure 10c) less or equal to one** is not necessarily an indication of an accurate retrieval…"

P22, L531-532: Is the multi-angle retrieval expected to retrieve smaller Reff? Can you explain that?

And why would smaller Reff lead to lower COT?

In case of mono-angular retrievals, high values of effective radius are retrieved, in particular in the cloudbow and in the glint regions. These effects are mitigated by multiangular retrieval. In case of muliangular retrievals, the effective radius tends thus to be smaller and more homogeneous over the scenes. A reduction of the effective radius leads to an increase of the backward scattering and, which results for the same visible radiance value, in a lower optical thickness.

We add, line 672  "A smallest effective radius leads to increase the backward scattering and so the reflected radiance, which results in a lower retrieved optical thickness"

P22, L533-534: This is not true. The measurement pair can be outside the 2D LUT spacelb (and I guess

this is what happens in the reported 5.9% cases of failed convergence).

We agree. We modify the sentence and add the possibility of having a radiance pair outside the LUT and the reference to the well-documented paper by Cho et al. (2015) regarding this issue. We add in the description of Figure 4, line 548:

"For the whole scene, failed retrievals account for 3.3% of the pixels. The failure may be associated with pairs of radiances outside the LUT that can occur for several reasons well documented in Cho et al. (2015)."

We delete the false assertion  "it is always possible to find a cloud model "  and write line 677: "Excepted in case of failed retrievals that occur for values outside the LUT ranges, the relatio between radiances and COT-Reff being monotonical, …"

P24, Fig. 12: The decrease in retrieval error from mono- to multi-angular retrievals is spectacular,

especially with respect to the vertical homogeneity and IPA assumptions. Can you explain in some

more detail how that is achieved? Still, differences between the two retrievals (Fig. 10 vs. Fig 4)

appear (much) larger than accommodated by the respective error estimates. Can you comment on

That?

Right, the differences in the assessment of the uncertainties due to the forward model are large. The reason that explains the difference between mono-angular and multi-angular measurements lies in the higher number of measurements used with the multi-angular retrieval. The state vector  retrieved with multi-angular measurements is less sensitive to the cloud model. We already discussed the advantages of multiangular retrieval in section 5 for example line 687 to 690 :

" On the other hand, multi-angular retrieval increases the constraint on the forward model make much more challenging  to find a solution allowing to fit the measurements.  The retrieved state is then consistent at the best with all the measurements associated with different viewing angles."

Concerning mean differences between mono and multi-angular retrieval, they are 1.18 and 6.48$\mu$m for optical thickness and effective radius respectively (news panels c and d in Figure 10) for mean values of 3.44 and 15.66$\mu$m. Even if not directly comparable, these values are in agreement with mean RSD values for COT and Reff, in Figure 12, which are 16 and 28% for optical thickness and 54% and 45% for effective radius. As added in the conclusion section (line 726-732), only numerical experiments, with known optical thickness and effective radius would allow to check if errors of the retrieved parameters are included in the uncertainties assessed by the method presented in the paper.

P24, Fig. 12: The mean Reff retrieval error due to measurement errors is 12.55 in Fig. 11 but 12 in
this figure, which is not consistent.
Right. This is a rounding error. We correct in the figure and indicates 13%
P25, L596: In Fig. 12 the mean COT error is 4%, not 5%.
Right this is 5% for the vertical profile error
P25, L605-607: Please remove since this was not shown (or alternatively include in the retrieval error
estimates).
We delete this sentence as we agree that we did not bring out the illumination and shadowing effects.

Technical corrections
P1, L22: Acronyms (POLDER in this case) must be written out.
Done
P1, L16/L17: '… without considering … the choice of ancillary data': What does it mean that the
choice of ancillary data is not considered?
The sentence  was deleted
P1, L31: 'uncertainties on': should be 'of'. Occurs frequently throughout. Please correct.
Thank you. It was corrected for the whole text
P2, L53: The second sentence does not follow from the first, so the word 'Therefore' is misplaced.
Deleted
P3, L76: increase -> increasing
Done
P3, L80: 'radiations' is not really a word.
Replaced by radiative energy
P3, L90: by its -> in
Done
P3, L96: vertical -> vertically
Done
P4, 113: Usually, the acronym is put between brackets after the full name instead of the other way

Round.

Done

P4, L124: Bayesian (with capital)

Done

P9, L242: Add lambda_a and lambda_b after wavelengths.

Done

P9, 239: Italic case is not needed here (similar occurrences throughout).

Modified

P9, 240: Variables in italic (R in R_eff should be italic). (similar occurrences throughout).

Done

P9, L243: (8) is duplicated.

Corrected

P9, L250: 'All the' -> 'the two'?

Done

P10, L271: 'implantation': do you mean 'implementation', 'inclusion', ..?

Inclusion

P10, L271: adjust -> adjusts

Done

P10, L306: 'measurement errors that cover the measurement errors'?

Keep just "based on 5% of measurement errors"

P11, L13: Italics appearing here and there are not needed and confusing.

Modified

P12, L327: Should (17) and (18) be reversed?

It is (19) and (20)

P12, L328: Should this be K_b_i instead of K_i?

Done

P12, L340: for -> to

Done

P14, Fig. 3: Minus sign in the x-axis label is confusing.

We change "-" by ";"

P15, L395: exctinction -> extinction

Done

P15, L417: minimized -> underestimated (?)

Done

First paragraph on page 16: here I give a more complete inventory of textual mistakes as guidance
for the rest of the manuscript.

P16, L422: Both bispectral and bi-spectral occur in the manuscript.

Bispectral is chosen

P16, L423: weak -> weakly

Done

P16, L423: .. channel partially absorbed by ..: how can a channel be absorbed?

We modified :"…and a radiance partially absorbed by the water droplets in the channel centered at 2200 nm,..."

P16, L424: on -> to

Done

P16, L424: Remove 'up to' (?) I guess all viewing angles are available. By the way, does this mean

that n_y = 13?

Some images in the sequence have to be removed because they were degraded. It leads to a decrease of the number of angles for the left side of the central image. In addition, the plane moves slightly above the cloud, which decreases the number of view directions in the edges. Consequently, 9 to 13 directions can be used for the retrieval depending of the part of the central images (see figure below)

[Figure]

Figure : Number of viewing angles used for the multiangular retrieval

P16, L425-426: 'This error is straightforward': how can an error be straightforward?

Straightforward was deleted

P16, L429: ertically -> vertically

Done

P17, L443: As noted before, do not write variables like COT, and mathematical operations like RSD,

in italics.

Done

P17, L457: 'enlarge the directions': what does that mean?

We modified "... and the bright surface, named glitter, is enlarged by waves formed by the wind."

P19, caption Fig. 7: 'model' missing after 'forward'?

Done

P19, L478: What are 'these differences'?

We detailed : "At high spatial resolution, these differences are mainly caused by the so-called smoothing effects that can increase or decrease the radiance according to the optical thickness gradient between the considered pixel and its neighbors."

P21, L503: assumption -> assumption

Done

P21, Fig. 10: For comparability with Fig. 4 it would be good to use the same color scales. Can you also add the mean values? Also, add some whitespace between the maps and the color bars.

As detailed above, the figure 10 was modified.

P22, caption Fig. 11: Add 'angle' after 'scattering'.

Done

P22, Fig. 11: Is this figure for the mono-angular retrieval?

yes , we now specify it in the legend

P23, L542: spatially -> spatial

Done

P23, L557: 'to the' is duplicated.

Corrected

P23, L557: what is a 'homogeneous assumption'?

We specified: "... the **cloud** homogeneous assumption **used** in the forward model".

P24, L571: 'retrieve' is duplicated.

corrected

P24, L583: horizontal -> horizontally, vertical -> vertically

Done

P25, L587: for -> to

Done

P25, L590: what is 'miss-knowledge'?

replaced  by unknown value

References: Journal names are missing in all references.

References

Platnick, S., Meyer, K. G., D., K. M., Wind, G., Amarasinghe, N., Marchant, B., Arnold, G. T., Zhang, Z.,

Hubanks, P. A., Holz, R. E., Yang, P., Ridgway, W. L., and Riedi, J., 2017: The MODIS Cloud Optical and

Microphysical Products: Collection 6 Updates and Examples From Terra and Aqua, IEEE T. Geosci.

Remote, 55, 502–525, doi: 10.1109/TGRS.2016.2610522.

Walther, A. and Heidinger, A. K., Implementation of the daytime cloud optical and microphysical

properties algorithm (DCOMP) in PATMOS-x, J. Appl. Meteorol. Climatol., 51, 1371-1390, doi:10.1175/JAMC-D-11-0108.1.